# Copy number variants as modifiers of breast cancer risk for *BRCA1/BRCA2* pathogenic variant carriers

The contribution of germline copy number variants (CNVs) to risk of developing cancer in individuals with pathogenic *BRCA1* or *BRCA2* variants remains relatively unknown. We conducted the largest genome-wide analysis of CNVs in 15,342 *BRCA1* and 10,740 *BRCA2* pathogenic variant carriers. We used these results to prioritise a candidate breast cancer risk-modifier gene for laboratory analysis and biological validation. Notably, the HR for deletions in *BRCA1* suggested an elevated breast cancer risk estimate (hazard ratio (HR) = 1.21), 95% confidence interval (95% CI = 1.09–1.35) compared with non-CNV pathogenic variants. In contrast, deletions overlapping *SULT1A1* suggested a decreased breast cancer risk (HR = 0.73, 95% CI 0.59-0.91) in *BRCA1* pathogenic variant carriers. Functional analyses of *SULT1A1* showed that reduced mRNA expression in pathogenic *BRCA1* variant cells was associated with reduced cellular proliferation and reduced DNA damage after treatment with DNA damaging agents. These data provide evidence that deleterious variants in *BRCA1* plus *SULT1A1* deletions contribute to variable breast cancer risk in *BRCA1* carriers.

Women who carry pathogenic variants in *BRCA1* (OMIM 113705) and *BRCA2* (OMIM 600185) have greatly increased risk of developing breast cancer. Recent cumulative risk estimates in high-risk families for developing breast cancer by age 80 years were 72% (95% confidence interval (95% CI, 65–79%)) for *BRCA1* and 69% (95% CI, 61–77%) for *BRCA2* pathogenic. variant carriers[1]. The significant variation of age at diagnosis of breast cancer between pathogenic variant carriers suggests additional factors, such as common inherited genetic variants, influence disease penetrance[2,3]. Large genome-wide association studies, facilitated by the Consortium of Investigators of Modifiers of *BRCA1/BRCA2* (CIMBA[2,3]), have demonstrated that >60 single nucleotide polymorphisms (SNPs) or small insertions or deletions (Indels) associated with cancer risk in the general population also are associated with breast cancer risk for *BRCA1/2* pathogenic variant carriers[4–6]. Moreover, population-based breast cancer polygenic risk scores are associated with modified breast cancer risk for carriers[7,8]. However, the identified single nucleotide variant modifiers account for <10% of heritable variation in risk in *BRCA1/2* pathogenic variant carriers[5].

Copy number variants (CNVs) cover between 5-10% of the human genome and, based on nucleotide coverage, are responsible for the majority of variation in the human genome[9,10]. CNVs exhibit substantial variability in both size and frequency and can disrupt gene function significantly by altering gene dosage, coding sequences, and gene regulation[11]. Germline CNVs overlapping the *BRCA1* and *BRCA2* gene loci are associated with the pathogenesis of breast cancer, accounting for <5% of known pathogenic variants in these genes (https://www.ncbi.nlm.nih.gov/clinvar). Multiple studies have utilised a genome-wide approach to identify associations between rare and common CNVs and the risk of developing breast cancer[12–16].

CNVs previously have been shown to be modifiers of hereditary breast cancer risk. In a genome-wide association study (GWAS) of CNVs in 2500 *BRCA1* pathogenic variant carriers, 52 gene loci were associated (unadjusted $p < 0.05$) with breast cancer risk[14]. Although no variant reached the widely-adopted genome-wide statistical significance threshold applied for SNP-centric GWAS ($p < 5 \times 10^{-8}$) and the study sample size was relatively small, the specific genes disrupted by CNVs had plausible biological consequences regarding cancer development. These data suggested that CNVs are an important modifier of hereditary breast cancer risk and highlighted the need for larger and more comprehensive CNV studies.

In this study, we conducted genome-wide CNV analyses of 15,342 *BRCA1* and 10,740 *BRCA2* pathogenic variant carriers using genotype data generated by the OncoArray Network[17]. We applied in silico and in vitro analyses to characterise a novel risk association between deletions overlapping *SULT1A1* (OMIM 171150) and decreased breast cancer risk for *BRCA1* pathogenic variant carriers.

## Results

**Copy number variants**. 857,647 CNVs (327,530 deletions and 530,117 duplications) were called in study participants, of which 374,210 CNVs (43.6%; 136,534 deletions and 237,676 duplications) overlapped at least one of 16,395 different gene regions. On average, each genome carried 14.4 CNVs (range = 0–63) that overlapped an average of 21.7 genes (range = 0–236). On average, duplications were detected nearly twice as often as deletions (9.1 (range = 0–55) versus 5.2 (range = 0–58), respectively) and affected twice as many gene regions (14.6 (range = 0–220) versus 7.0 (range = 0–207), respectively).

**Evaluation of CNV calling**. The sensitivity and specificity of our CNV calling was assessed by comparing diagnostically identified *BRCA1* and *BRCA2* CNVs to the CNVs called by our analysis. In our cohort, 1,138 *BRCA1* and 166 *BRCA2* diagnostically identified CNVs overlapped five or more probes and passed our variant filtering; we detected 678 and 155 for *BRCA1* and *BRCA2*, respectively. Furthermore, our genome-wide analysis called 851 *BRCA1* and 183 *BRCA2* CNVs, of which 151 and 28 CNVs were not supported by diagnostic testing of *BRCA1* and *BRCA2*, respectively. Together, our CNV calling achieved an 82.2% and 84.7% detection specificity and 59.6% and 92.8% detection sensitivity for CNVs in *BRCA1* and *BRCA2*, respectively.

Separate analysis of deletions and duplications found that PennCNV performed better with deletion calling verses duplication calling (Supplementary Data 1). The sensitivity of calling *BRCA1* and *BRCA2* deletions were 84.1% and 91.3%, respectively, while the sensitivity of calling *BRCA1* and *BRCA2* duplications were 68.6% and 39.1%, respectively. Similarly, the specificity of calling *BRCA1* and *BRCA2* deletions was 70.4% and 94.8% and duplications was 25.0% and 60.0%, respectively. A review of the diagnostic CNVs not called by PennCNV found that the majority of uncalled CNVs overlapped were the same variants. For example, 54.1% of *BRCA1* deletions not detected by PennCNV were the same variant (c.5333-36_5406 + 400del510), which was only successfully called by PennCNV for 1.4% of the variant carriers.

Genome-wide CNV calling was assessed further for three cases using whole-genome sequencing (WGS). In the three whole-genome sequenced cases, 4444 (Case 1), 4540 (Case 2), and 4545 (Case 3) CNV calls passed confidence filtering, of which 1884 (Case 1), 1981 (Case 2), and 1940 (Case 3) overlapped gene regions, respectively (Supplementary Data 2). A total of 10 of 14 (71.4%—Case 1), 23 of 42 (54.8%—Case 2), and 13 of 19 (68.4% —Case 3) of PennCNV calls were supported by a CNV called in the WGS data (Supplementary Data 3). Of the CNVs not supported by WGS, 4 of 4 (100%—Case 1), 12 of 19 (63.2%—Case 2), and 6 of 6 (100%—Case 3), CNVs were supported by a previously-published CNV map[10]. All duplications that were not supported by WGS were supported by the CNV map while approximately half the deletions were not (Supplementary Data 4). Each of the three cases also carried a diagnostically identified pathogenic *BRCA1* deletion that was called by WGS CNV calling. Together, these data provide confidence that >80% of deletions and duplications called by PennCNV in this study cohort appear to be true calls.

**Prioritization of candidate breast cancer CNV risk loci**. To prioritise genes for in silico and functional analyses, we selected candidate gene loci with $p < 0.01$ from retrospective likelihood analysis, effectively restricting hazard ratios to >1.25 and <0.75 (Supplementary Data 5–8). Putative CNVs at 31 gene regions passed this threshold. For 16 of these 31 regions, the proportion of unique CNVs represented in a published human CNV map[10] was <95%. Although none of the CNV regions passed significance thresholds when adjusted for multiple hypothesis testing (See Methods; deletions in *BRCA1* carriers—$p \leq 8 \times 10^{-6}$; duplications in *BRCA1* carriers—$p \leq 5 \times 10^{-6}$; deletions in *BRCA2* carriers— $p \leq 1 \times 10^{-5}$; and duplications in *BRCA2* carriers—$p \leq 6 \times 10^{-6}$), we used these results to prioritise a candidate risk-modifier gene for laboratory analysis and biological validation.

Deletions overlapping *BRCA1* increased breast cancer risk (hazard ratio (HR) = 1.29, 95%CI = 1.13–1.49, $p = 1.98 \times 10^{-4}$) (Supplementary Data 5) for *BRCA1* pathogenic variant carriers. This result was explored further as the analysis did not directly compare the effect of *BRCA1* deletions and *BRCA1* non-deletion pathogenic variants. Clinically diagnosed variants for *BRCA1* and *BRCA2* carriers were categorised by type (deletions, duplications, and small

**Table 1 Breast cancer hazard ratio estimates using a single model comparing other *BRCA1/2* pathogenic variants versus (1) deletions, and (2) duplications[a].**

|  | Mutation type | Unaffected | Affected | HR | 95%CI | | *p* |
|---|---|---|---|---|---|---|---|
| ***BRCA1* carriers** | Other | 8669 | 8483 | 1.00 | [reference] | | |
| | Deletions | 636 | 789 | 1.21 | 1.09 | 1.35 | 4.35E-04 |
| | Duplications | 153 | 192 | 1.21 | 0.99 | 1.48 | 6.60E-02 |
| ***BRCA2* carriers** | Other | 5913 | 6204 | 1.00 | [reference] | | |
| | Deletions | 85 | 111 | 1.11 | 0.79 | 1.54 | 5.54E-01 |
| | Duplications | 6 | 16 | 1.52 | 0.61 | 3.77 | 3.68E-01 |

[a] Weighted cohort models fitted separately for *BRCA1* and *BRCA2* carriers. Weights were calculated assuming *BRCA1/2* carrier breast cancer incidences from most recent age cohort in Antoniou et al., (2008). Models were stratified by country and Ashkenazi Jewish ancestry, and adjusted for birth cohort and genotyping array. Cluster robust variances were estimated using families as clusters. *HR* hazard ratio, *95%CI* 95% confidence interval; *p*-value.

**Table 2 Putative copy number variants assessed by TaqMan assays.**

| Gene symbol | CNV type | Variant carrier | Assay ID | Overlap CNV map[1] | Proportion validated |
|---|---|---|---|---|---|
| ***SULT1A1*** | Deletion | *BRCA1* | 1 | Yes | 100% (8/8) |
| ***TERT*** | Deletion | *BRCA2* | 1 | No | 0% (0/1) |
| | | | 2 | No | 0% (0/1) |
| | | | 3 | No | 33% (1/3) |
| ***LSP1*** | Duplication | *BRCA2* | 1 | No | 50% (1/2) |

*CNV* copy number variant. [1]Zarrei et al (2015) stringent CNV map.

variants [i.e. nonsense, missense, frame shift, Indel, and splice site]). Assessing the HRs for CNV *versus* non-CNV pathogenic variants, separately for *BRCA1* and *BRCA2* suggested elevated breast cancer risk for *BRCA1* deletions (HR = 1.21, 95%CI = 1.09–1.35) but not *BRCA2* deletions (Table 1, Supplementary Data 9). These results remained similar after excluding missense variant carriers from the analysis (Supplementary Data 9). Estimated HRs were elevated for duplications versus non-duplication pathogenic variants (deletions were excluded) for *BRCA1* duplication carriers (HR = 1.21, 95% CI = 0.99–1.48; *p* = 0.066), and *BRCA2* duplication carriers (HR = 1.52, 95%CI = 0.61–3.77, *p* = 0.39); however, results for *BRCA2* were less definitive given the smaller sample size and wide confidence intervals.

Putative duplications overlapping the breast cancer tumour suppressor gene *STK11* suggested decreased risk of breast cancer in our study for both *BRCA1* carriers (HR = 0.49, 95%CI 0.29–0.81, *p* = $5.4 \times 10^{-3}$) and *BRCA2* carriers (HR = 0.44, 95% CI 0.22–0.88, *p* = $9.2 \times 10^{-3}$). Putative deletions overlapping *TERT* and duplications overlapping *LSP1*, two loci previously shown to be associated with breast cancer risk for *BRCA1* (*TERT* locus) and *BRCA2* (*TERT* and *LSP1* loci) pathogenic variant carriers from SNP-based studies[18,19], suggested increased risk (HR = 1.92, 95%CI = 1.06–3.46, *p* = $6.0 \times 10^{-3}$) and decreased risk (HR = 0.13, 95%CI = 0.04–0.45, *p* = $3.3 \times 10^{-3}$) breast cancer risk for *BRCA2* pathogenic variant carriers in this study, respectively. However, analysis of *TERT* and *LSP1* CNVs using TaqMan assays with available DNA showed only 25% (1/4) of predicted deletions overlapping *TERT* and 50% (1/2) predicted duplications overlapping *LSP1* were successfully validated (Table 2, Supplementary Fig. 1). These results are consistent with the observation that predicted CNVs overlapping *LSP1* and *TERT* were not found in a published genomic map of human CNVs[10].

**Identification of *SULT1A1* as a candidate modifier gene.** CNV loci suggested to modify breast cancer risk estimates in *BRCA1/2* pathogenic variant carriers, were examined to identify a candidate gene for functional characterisation using in silico and in vitro

assays. *SULT1A1* (sulfotransferase 1A1) was selected as a novel candidate modifier based on potential biological mechanisms of action and because overlapping CNVs had a population frequency above 1%.

In our study, CNV deletions overlapping *SULT1A1* were identified in 1.7% of *BRCA1* pathogenic variant carriers and they suggested a decreased breast cancer HR (HR = 0.73, 95% CI = 0.59–0.91, *p* = $9.1 \times 10^{-3}$). Deletions overlapping all eight *SULT1A1* exons of the reference transcript were validated in all eight available DNA samples using both TaqMan assays and multiplex ligation-dependent probe amplification (MLPA; Supplementary Fig. 2), and were identified in the CNV map published by Zarrei et al (Supplementary Fig. 1c). Furthermore, CNVs involving the *SULT1A1* gene locus had an expression dosage effect in breast tumours (Fig. 1).

**Characterisation of *SULT1A1* knockdown in *BRCA1*[+/−] cells.** To model the effect of *SULT1A1* deletions in *BRCA1* carriers, a pair of isogenic MCF7 breast cell lines with and without a pathogenic variant (*BRCA1* c.2432_2433del) were created using CRISPR-Cas9 (Fig. 2a). A comparison of the isogenic control (MCF7–C1) and pathogenic variant carrying cells (MCF7–*BRCA1*[+/−]) with the parent MCF7 cells (MCF7–WT) showed no significant difference in cell proliferation (Fig. 2b, Supplementary Data 10). There were also no differences in the relative expression of BRCA1 mRNA between the isogenic MCF7–C1 and MCF7–*BRCA1*[+/−] lines and the parent MCF7–WT line (Fig. 2c, Supplementary Data 10). However, the MCF7–*BRCA1*[+/−] cells showed a significant 25% (*p* = $4.44 \times 10^{-4}$) reduction in ESR1 expression (Fig. 2d, Supplementary Data 10) and a significant 78% (*p* = $2.03 \times 10^{-3}$) increase in CYP1A1 expression (Fig. 2e, Supplementary Data 10), consistent with breast cells with a pathogenic *BRCA1* variant[20]. There was no significant difference in *SULT1A1* mRNA expression between the MCF7–WT and the isogenic MCF7–C1 and MCF7–*BRCA1*[+/−] cells (Fig. 2f, Supplementary Data 10).

siRNA was used to transiently reduce the relative expression of *SULT1A1* mRNA in the isogenic MCF7–C1 and MCF7–*BRCA1*[+/−]

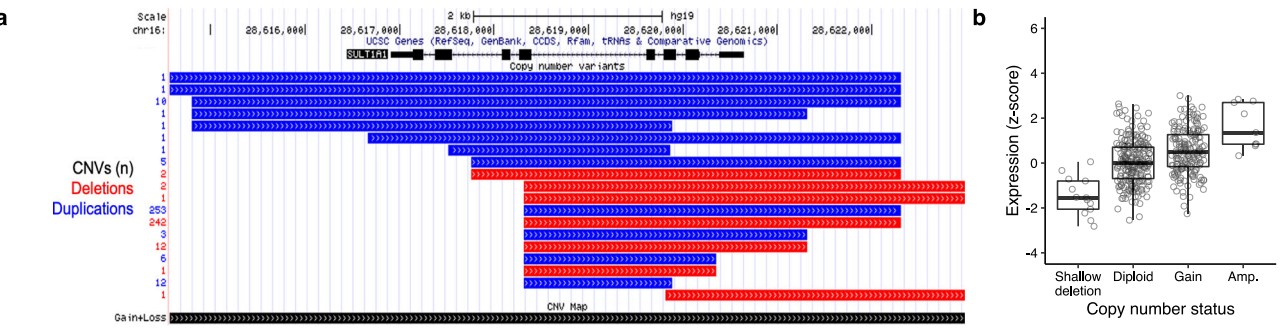

**Fig. 1 Characterisation of SULT1A1. a** Genomic viewer (UCSC) of the SULT1A1 gene locus with copy number variants; deletion (red) and duplication (blue). **b** Dosage effect in breast tumours with SULT1A1 copy number variants. CNVs copy number variants.

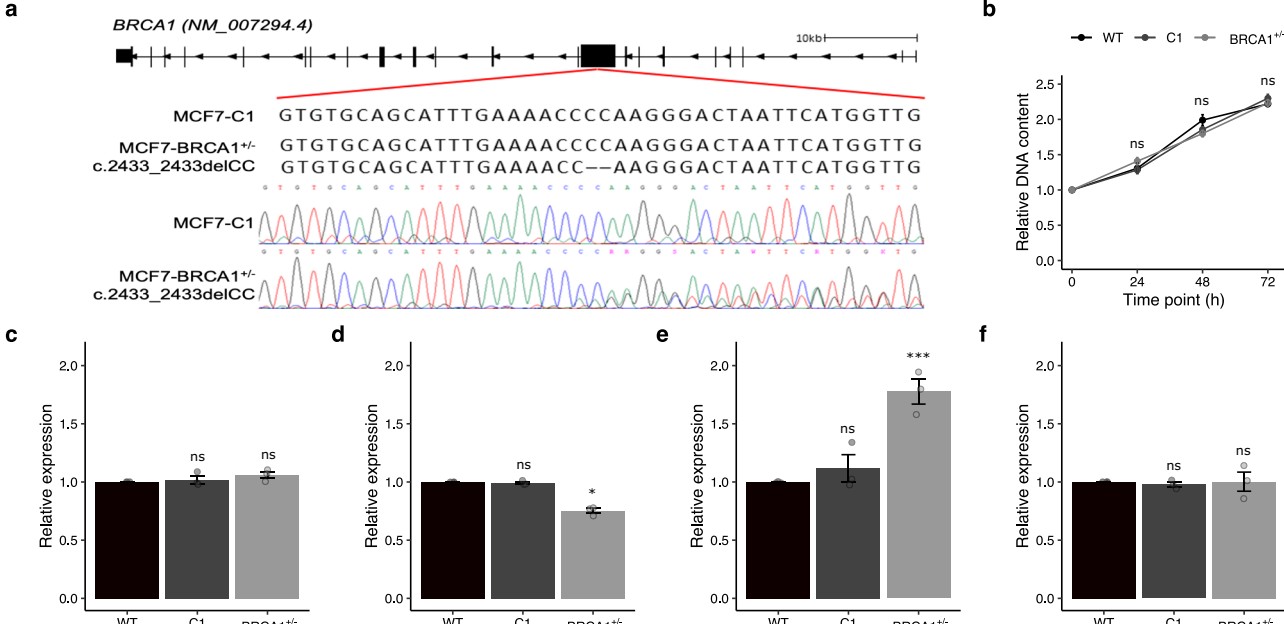

**Fig. 2 Characterisation of MCF7–WT, and isogenic MCF7–C1, and MCF7–BRCA1$^{+/-}$ cell lines. a** Sequence of heterozygous pathogenic BRCA1 c.2432_2433del variant introduced by CRISPR-Cas9. **b** Relative proliferation of MCF7–WT and clonally expanded CRISPR-Cas9 MCF7–C1 and MCF7–BRCA1$^{+/-}$ cells for 72 h post seeding. Relative expression of **c** BRCA1, **d** ESR1, **e** CYP1A1, and **f** SULT1A1 for MCF7–WT, MCF7–C1 and MCF7–BRCA1$^{+/-}$ cells. 4-OHE2 4-hydroxyestradiol, MMC Mitomycin C, Error bars = standard error of the mean; ns = $p > 0.05$; *$p < 0.05$; **$p < 0.01$; ***$p < 0.001$; $n = 3$ independent biological replicates; unpaired two-sided $t$-test.

lines. Compared with the non-targeting siRNA control, the relative expression of *SULT1A1* was approximately half in both isogenic lines 72 h after transfection targeting *SULT1A1* (Fig. 3a, Supplementary Data 11). As deletions overlapping *SULT1A1* were suggested to decrease breast cancer risk in *BRCA1* pathogenic variant carriers, the relative expression of *BRCA1* was quantified to assess if the *SULT1A1* knockdown affected its regulation (Fig. 3b, Supplementary Data 11). However, there was no significant change in the *BRCA1* expression for the *SULT1A1* knockdown cells compared with the transfection control for either of the MCF7–C1 or the MCF7–*BRCA1*$^{+/-}$ lines.

The proliferation of the transfected MCF7–C1 and MCF7–*BRCA1*$^{+/-}$ cells was assessed by measuring relative DNA content. Knockdown of *SULT1A1* expression did not alter the proliferation of the MCF7–C1 compared with the transfection control (Fig. 3c, Supplementary Data 11). However, when *SULT1A1* expression was reduced in MCF7–*BRCA1*$^{+/-}$ cells, there was a 14% ($p = 1.17 \times 10^{-2}$) decrease in proliferation compared with the transfection control after 72 h of growth (Fig. 3d, Supplementary Data 11).

Because BRCA1-deficient cells are hypersensitive to DNA damaging agents and have an impaired DNA damage repair

response[20,21], we investigated whether knockdown of *SULT1A1* expression altered the amount of damage caused by the DNA damaging agents 4-hydroxyestradiol (4-OHE2) and Mitomycin C (MMC) using the well-established comet and γH2AX and 53BP1 immunostaining assays. Analysis using comet assays, showed that treatment with 4-OHE2 ($F(1, 8) = 5.79$, $p = 4.28 \times 10^{-2}$; Fig. 3e, Supplementary Data 11) and MMC ($F(1, 8) = 5.73$, $p = 4.36 \times 10^{-2}$; Fig. 3f, Supplementary Data 11) increased comet tail moment length in MCF7–*BRCA1*$^{+/-}$ cells. There also was evidence that transfection with siSULT1A1 reduced the average comet tail moment length for both 4-OHE2 ($F(1, 8) = 9.76$, $p = 1.41 \times 10^{-2}$; Fig. 3e, Supplementary Data 11) and MMC ($F(1, 8) = 5.69$, $p = 4.43 \times 10^{-2}$; Fig. 3f, Supplementary Data 11) treated MCF7–*BRCA1*$^{+/-}$ cells. Assessing the effect of *SULT1A1* knockdown in MCF7–*BRCA1*$^{+/-}$ and MCF7–C1 cells using the γ-H2AX and 53BP1 immunostaining assay gave rise to analogous results (Fig. 3e–h). Although there was evidence that treatment with 4-OHE2 increased the average number of co-localised DNA damage foci ($F(1, 8) = 24.36$, $p = 1.14 \times 10^{-3}$; Fig. 3g, Supplementary Data 11) in MCF7–*BRCA1*$^{+/-}$ cells, the reduction in the number of foci caused by the siSULT1A1 transfection did not

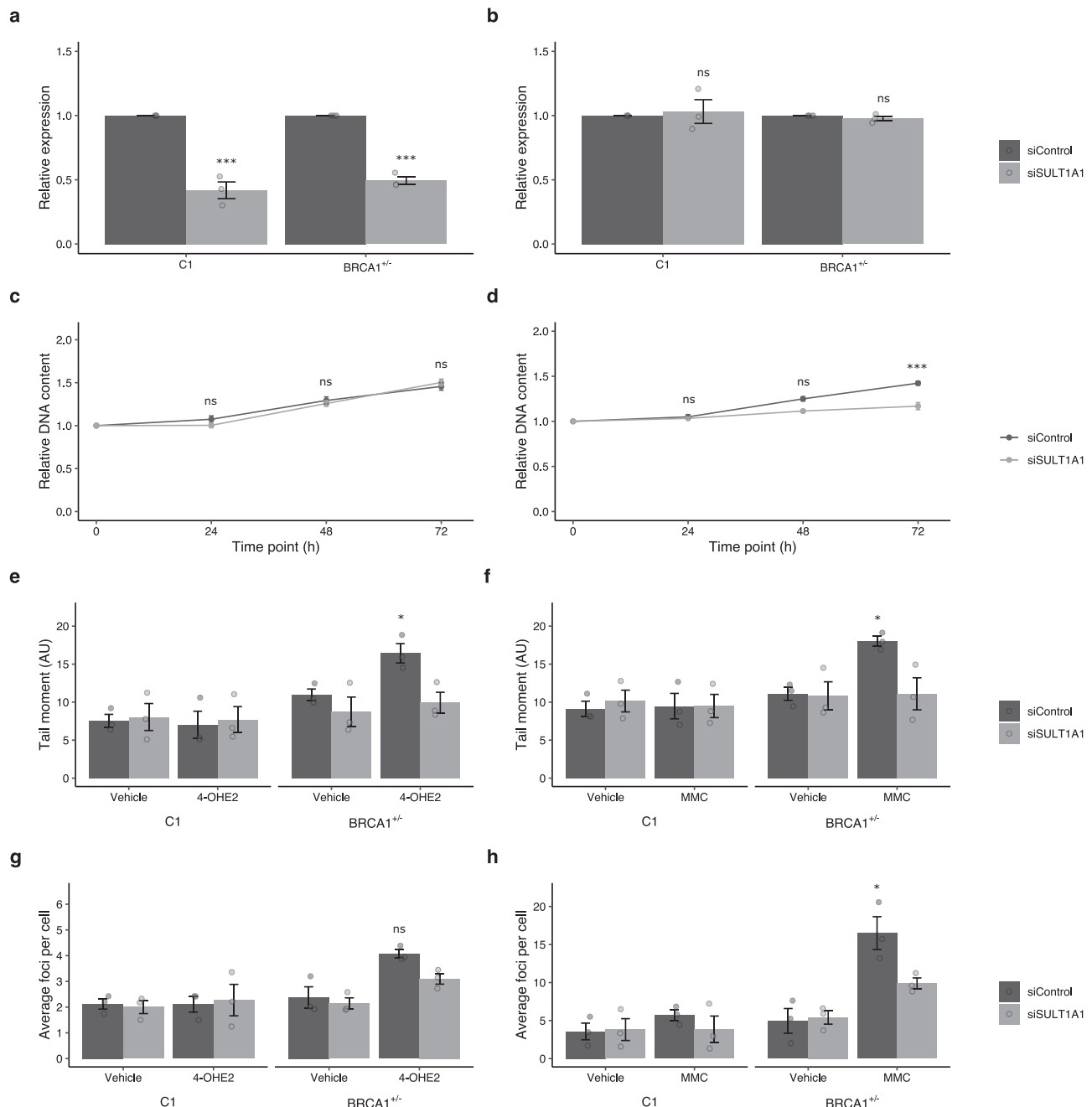

**Fig. 3 siRNA knockdown of SULT1A1 reduces proliferation and DNA damage of MCF7–BRCA1$^{+/-}$ cells but not MCF7–C1 cells.** Expression of (**a**) SULT1A1 and (**b**) BRCA1 72 h post transfection. Relative DNA content of transfected (**c**) MCF7–C1 and (**d**) MCF7–BRCA1$^{+/-}$ cells 72 h post transfection. Quantification of transfected DNA damage using the comet assay (**e**, **f**) and ϒ-H2AX/53BP1 foci quantification (**g**, **h**) for MCF7–C1 and MCF7–BRCA1$^{+/-}$ cells, with and without siSULT1A1 transfection, 21 h post treatment with 1 μm 4-OHE2 (**e** and **g**) or 10 μm MMC (**f** and **h**). Significance of differences in relative expression and DNA content was determined by unpaired two-sided *t*-test. Gene expression and DNA content of siSULT1A1 transfected cells normalised to siControl transfected cells. Differences in DNA damage were determined by two-way analysis of variance. Example images of comet and ϒ-H2AX/53BP1 foci shown in Supplementary Figs. 3–6. 4-OHE2 = 4-hydroxyestradiol; MMC = Mitomycin C; Error bars = standard error of the mean; ns = $p > 0.05$; *$p < 0.05$; **$p < 0.01$; ***$p < 0.001$; $n = 3$ independent biological replicates.

reach statistical significance ($F(1, 8) = 5.02$, $p = 5.5 \times 10^{-2}$). Additionally, for the MMC treated MCF7–BRCA1$^{+/-}$ cells there was a significant interaction between the siSULT1A1 transfection and MMC treatment ($F(1, 8) = 5.80$, $p = 4.3 \times 10^{-2}$; Fig. 3h, Supplementary Data 11). This effect was significant for MMC treated MCF7–BRCA1$^{+/-}$ cells ($F(1, 4) = 8.44$, $p = 4.4 \times 10^{-2}$) but not the vehicle control ($F(1, 4) = 0.06$, $p = 8.19 \times 10^{-1}$). There was no evidence that siRNA transfection or drug

treatments affected the average comet tail moment or the number of γ-H2AX and 53BP1 foci for the MCF7–C1 cells. Example images of comet and γ-H2AX and 53BP1 immunostaining assay are shown in Supplementary Figs. 3–6.

## Discussion
Germline CNVs are an important source of genetic variation that have previously been understudied in relation to breast and

ovarian cancer risk. Here, we have conducted the largest and most comprehensive genome-wide association study of CNVs and breast cancer risk for *BRCA1* and *BRCA2* pathogenic variant carriers. We identified putative CNVs in up to 31 putative gene regions that were associated (unadjusted $P < 0.01$) with breast cancer risk for *BRCA1/2* pathogenic variant carriers, with CNVs at 15 of these regions present in a human CNV map[10]. Although none of the CNV regions passed significance thresholds when adjusted for multiple hypothesis testing, we used these results to prioritise a candidate risk-modifier gene for laboratory analysis and biological validation. Consistent with observations from the human CNV map, we validated positive CNV calls overlapping the *SULT1A1* gene, and revealed false positive CNV calls at two candidate modifier gene regions (*LSP1* and *TERT*). CNV deletions overlapping the lead candidate modifier *SULT1A1* showed decreased breast cancer risk in *BRCA1* pathogenic variant carriers. In silico analysis of *SULT1A1* suggested that deletions overlapping this gene leads to reduced expression. In vitro analyses showed that reduced *SULT1A1* expression in cells carrying a heterozygous *BRCA1* pathogenic variant led to reduced cellular proliferation and reduced DNA damage after treatment with DNA damaging agents.

Both SNP and CNV variants at the *SULT1A1* locus have previously been shown to be associated with SULT1A1 enzymatic activity. The common *SULT1A1* p.(Arg213His) (rs9282861) polymorphism leading to the SULT1A1*2 variant has been examined in a series of functional and association studies. SULT1A1*2 has a two-fold lower catalytic activity and stability than its high-activity p.Arg213 counterpart (SULT1A1*1), and has been associated with increased cancer risk in multiple tissue types[22–25]. Studies examining the association between the rs9282861 polymorphism and breast cancer risk have yielded inconsistent results showing an increase in risk in some studies[26–29] and no association in others[30–32]. The OncoArray probe for rs9282861 failed quality control, therefore no genotype data was available. Interestingly, the rs200802208 Indel located near *SULT1A1* was imputed from these arrays and analyses showed that the del allele was associated with decreased risk of breast cancer in *BRCA1* pathogenic variant carriers (HR = 0.48, 95%CI 0.29–0.79, $p = 4.3 \times 10^{-3}$)[5]. rs9282861 is not in 1000 G reference panel, therefore it is unknown whether this SNP exists in linkage disequilibrium with rs200802208. CNVs overlapping *SULT1A1* are strongly associated with SULT1A1 activity and explain more of the observed in vitro variability in SULT1A1 activity than SNPs, with activity proportional to *SULT1A1* copy number[33–36]. Individuals who are homozygous null for *SULT1A1* do not present with any overt phenotypes[37]. This finding corresponds with results from phenotypic analyses of mouse *SULT1A1* knockouts which are viable, and which also lack any outward phenotype[38,39]. However, the absence of functional *SULT1A1* enzyme in mouse knockouts has been reported to reduce the number of DNA adducts caused by DNA damaging agents that are converted to mutagenic metabolites by SULT1A1[39].

The mechanism by which CNV deletions overlapping *SULT1A1* were associated with lower *BRCA1*-associated breast cancer risk may be linked to the production of potentially toxic catechol oestrogens by the Cytochromes P450 (CYP) enzymes. SULT1A1 is an important SULT isoform that is expressed widely in human tissues and plays an important role in the metabolism, bioactivation, and detoxification of carcinogens, medications, and steroid hormones[33,34]. *SULT1A1* has established germline common CNVs and SNPs that are known to alter its activity[40]. *SULT1A1* is most abundantly expressed in the liver, but is also expressed in the brain, breast, intestine, and endometrium[41–44]. *SULT1A1* expression is related to disease state, with plentiful expression in most breast tumours[42,45]. In normal breast cells,

BRCA1 regulates oestrogen metabolism and metabolite-mediated DNA damage by repressing transcription of the oestrogen-metabolising enzyme CYP1A1[20,46]. However, levels of CYP1A1 are higher in breast cells lacking *BRCA1* function and promote the formation of the carcinogenic 2-hydroxyestradiol (2-OHE2)[20]. Further metabolism of 2-OHE2 by catechol O-methyltransferase (COMT) is thought to have a risk reducing effect by catalysing the formation of 2-methoxyestradiol (2-MeOE2), a metabolite which interacts with the tubulin colchicine-binding site during polymerisation and which has anticarcinogenic effects by suppressing cell proliferation. In turn, SULT1A1 is an efficient catalyst of 2-MeOE2 sulfation producing 2-MeOE2-3S, a sulfate conjugate with diminished activity[33,47]. It is possible that the decreased risk associated with *SULT1A1* deletions for *BRCA1* pathogenic variant carriers and the decrease in cell proliferation and amount of DNA damage for MCF7–BRCA1$^{+/-}$ with a SULT1A1 knockdown cells may both be linked to 2-MeOE2 abundance. That is, reduced SULT1A1 activity promotes the accumulation of 2-MeOE2 and slows the proliferation of breast cells with unbalanced E2 metabolism. Indeed, the SULT1A1 substrate, 2-MeOE2, has previously been proposed as a potential preventative agent for breast cancer[48]. A similar relationship between CYP1A1 and SULT1A1 activity and reduced breast cancer risk has been demonstrated previously. In a study of pairwise combinations of oestrogen metabolism alleles and breast cancer risk, the *SULT1A1*2* genotype was assessed in combination with a *CYP1A1* missense variant (*CYP1A1*2 C*) that has increased inducibility to produce catechol oestrogens[49]. For European-Americans, carrying the *CYP1A1*2 C* genotype was associated with increased breast cancer risk (odds ratio (OR) = 1.71, 95%CI = 1.09–2.67). However, carrying the *CYP1A1*2 C* allele in combination with a *SULT1A1*2* allele was strongly protective against developing breast cancer (OR = 0.14, 95% CI = 0.04–0.56) compared with women carrying only the *CYP1A1*2 C* allele. There was no association between the *CYP1A1*2 C* (rs1048943) polymorphism and breast cancer risk in *BRCA1* pathogenic variant carriers[5]. These results further suggest that the balance between the generation of catecholestrogens and catecholestrogen sulfation may be an important mechanism for modulating breast cancer risk and worthy of future investigation.

Our study provides strong evidence that deletions overlapping *BRCA1* are associated with a 1.21-fold higher risk of developing breast cancer. Large deletions in *BRCA1* have previously been shown to be associated with an increased risk of breast cancer risk (OR = 1.42) compared with carriers of *BRCA1* pathogenic single nucleotide variants or Indels[50]. Similarly, a series of studies have reported a higher incidence of CNVs in both *BRCA1* and *BRCA2* when cases have a family history that includes high-risk features, such as early-onset disease[51,52]. Although a mechanism that explains the higher risk for *BRCA1* deletion carriers is unclear, one possible explanation is that large genomic variants disrupt key BRCA1 domains or cause nonsense-mediated mRNA decay, whereas some single nucleotide variants in *BRCA1* avoid nonsense-mediated decay and retain partial function. For example, the p.(Arg1699Gln) variant in *BRCA1* produces a protein with ambiguous behaviour in a variety of functional assays and is also associated with an intermediate risk[53,54]. Furthermore, variants in both *BRCA1* and *BRCA2* are proposed to have variant-specific risks, that coincide with known or hypothesised functional domains and vary by variant type and location[1,55]. Our results support the hypothesis that breast cancer risk for women carrying large deletions in *BRCA1* is greater than those pathogenic single nucleotide variants or Indels, which may have implications for clinical risk assessment and management of CNV carriers.

Despite this being the largest sample size of *BRCA1* and *BRCA2* pathogenic variant carriers available to date, the low frequency of CNVs results in limited power for detecting significant associations after adjusting for multiple comparisons. As a result, a nominal screening threshold of 0.01 was used which is arbitrary and is therefore a limitation of the study. Nevertheless, this is the largest extant dataset available for examining genetic modifiers of *BRCA1* and *BRCA2* related risk. While larger studies, such as the new Confluence project (https://dceg.cancer.gov/research/cancer-types/breast-cancer/confluence-project), may lead to improved statistical power to detect CNV associations, evaluating uncommon genetic variation such as CNVs that overlap *SULT1A1* and other potential modifier genes in *BRCA1/2* pathogenic variant carriers will remain a challenge. Furthermore, CNV calling algorithms have limitations which lead to false CNV calls, thus highlighting the importance of using ancillary data to prioritise regions for downstream analyses. Here we show that functional analysis of a candidate modifier gene using a model cell line is able to provide additional evidence that *SULT1A1* deletions lead to reduced risk of breast cancer in *BRCA1* pathogenic variant carriers. If verified, future therapeutic intervention studies targeting SULT1A1 in *BRCA1* pathogenic variant carriers may lead to new medical options for reducing breast cancer risk.

In conclusion, our study provides evidence that CNVs contribute to the variability in breast cancer risk among *BRCA1* and *BRCA2* pathogenic variant carriers. Characterising pathogenic variant type in *BRCA1*, and future screening for deletions overlapping *SULT1A1*, may produce variables to be incorporated with other modifying factors to develop a more comprehensive model of breast cancer risk. For example, integrating these genetic data into the CanRisk Web Tool (https://www.canrisk.org/)[56] along with family history, lifestyle/hormonal risk factors, common genetic susceptibility variants, and mammographic density, may further improve breast cancer risk predictions. Such a model may better inform patient decisions regarding breast cancer risk management.

## Methods

**Study cohort.** Female *BRCA1* and *BRCA2* pathogenic variant carriers were from study centres across North America, Europe, and Australia participating in CIMBA (Supplementary Data 12), as reported previously[4,5]. Eligibility criteria for study participants included: (1) female carriers of *BRCA1* or *BRCA2* pathogenic variants; and (2) minimum 18 years of age at recruitment. A complete list of *BRCA1* and *BRCA2* pathogenic variants are deposited in the ClinVar database (https://www.ncbi.nlm.nih.gov/clinvar/submitters/505954/) and a filtered list of those in participants that were analysed in this study (post-quality control) is shown in Supplementary Data 13. There were 7725 (50.4%) *BRCA1* and 5488 (51.1%) *BRCA2* pathogenic variant carriers diagnosed with breast cancer (Supplementary Data 14). All participants were recruited for research studies using ethically approved protocols at host institutions.

**CNV detection and quality control.** DNA samples were genotyped using the OncoArray-500k BeadChip (Illumina) with 533,631 probes, and standard sample quality control exclusions were performed as previously described for the SNP genotype analysis[17]. GenomeStudio (Illumina) was used to export Log R Ratios (LRRs) and B allele frequencies (BAFs) for each sample as previous described[14]. A principal components adjustment (PCA) was run on the LRR to remove noise using the bigpca package (V1.1)[57] in the statistical platform R (V3.5.2)[58]. CNV calls were generated using PennCNV[59]. Probes that failed to cluster using Illumina's Gentrain algorithm ($n = 4857$) and probes on the Y chromosome were removed from these results. Neighbouring CNVs with a gap of <20% of the total length of the combined CNVs, were merged using the PennCNV clean_cnv.pl script. For the current study we determined genetic ancestry using a principal components approach described elsewhere[5]. A total of 15679 *BRCA1* and 10981 *BRCA2* pathogenic variant carriers of European ancestry were assessed.

The study cohort was filtered to remove samples that failed study requirements or quality controls (Supplementary Fig. 7). Samples were removed if they met criteria listed in Supplementary Fig. 7, or if they met the following criteria: PennCNV measures of LRR standard deviation (s.d.) > 0.28, BAF drift > 0.01, waviness factor deviating from 0 by >0.05; LRR outliers > 0.1, BAF s.d. ≥ 0.2, LRR s.d. ≥ 0.4. Additionally, samples with >100 CNVs were excluded. To reduce false

positive calls, only copy number variants called by five or more probes were retained for analysis. A total of 857,647 CNVs carried by 15,342 *BRCA1* and 10,740 *BRCA2* pathogenic variant carriers passed quality control steps and were assessed.

**Defining gene-centric CNVs.** To identify genomic loci that influence breast cancer risk for *BRCA1* and *BRCA2* pathogenic variant carriers, a non-redundant gene-centric approach was used. Gene regions were derived from the University of California, Santa Cruz (UCSC) GRCh37/Hg19 gene track (updated: 14 June 2013) and were restricted to chromosomes 1–22, and chromosome X. In total, 30,336 gene regions with 27,038 unique gene symbols were derived and used in our analysis. CNVs that overlapped a gene region by one or more base pairs were identified in a genome-wide scan in R (V3.5.3) using the GenomicRanges package (V1.4)[60]. Overall, 374,210 CNVs overlapped one or more of 16,395 unique gene regions and were retained for statistical analysis.

**Breast cancer risk association analysis.** The association analyses between breast cancer risk and copy number deletions and duplications were conducted separately for *BRCA1* and *BRCA2* pathogenic variant carriers. Study participants were followed from birth until the age at first breast cancer diagnosis, age at ovarian cancer diagnosis or bilateral prophylactic mastectomy (whichever occurred first), or at the age at last observation. Only those diagnosed with breast cancer were considered to be affected. Pathogenic variant carriers with ovarian cancer were considered unaffected, and censored at ovarian cancer diagnosis.

*BRCA1* and *BRCA2* pathogenic variant carriers were sampled non-randomly with respect to their disease status. Therefore, to evaluate associations between deletions and duplications and breast cancer risk, we analysed these data using a kinship adjusted score test based on the retrospective likelihood of observing the CNV conditional on the observed phenotype to account for the non-random ascertainment[61]. This model is stratified by country and Ashkenazi Jewish ancestry but is unable to adjust for covariates. An approximation method yields HR and 95%CI estimates based on this score test[61]. Instances in which a non-overlapping deletion and duplication was called in the same gene region were excluded, however this occurrence was relatively uncommon (<1% of participants were removed after the analysis of 99.3% of gene regions). Retrospective likelihood analysis of variants was performed using R (V3.3.1) and bespoke software (available on request). Conservative significance thresholds were based on the number of effective tests in this gene-centric CNV study. After excluding gene regions with no overlapping CNVs, thresholds were as follows: deletions in *BRCA1* carriers—$p \le 0.05/6551 = 8 \times 10^{-6}$; duplications in *BRCA1* carriers—$p \le 0.05/10240 = 5 \times 10^{-6}$; deletions in *BRCA2* carriers—$p \le 0.05/5094 = 1 \times 10^{-5}$; and duplications in *BRCA2* carriers—$p \le 0.05/8469 = 6 \times 10^{-6}$. Hypervariable regions of the genome that are prone to CNV calling errors, including the human leucocyte antigens, immunoglobulin superfamily, and olfactory receptor genes were excluded from the final gene lists.

Models to estimate the associations (HRs) of deletions and duplications simultaneously took the form of a weighted cohort analysis[61,62]. This method assigns different weights to unaffected and affected carriers depending on their age at diagnosis/censoring such that the weighted cohort mimics a true cohort. Weights were calculated using the most recent birth cohort incidence estimates from Antoniou et al.[18]. These models were stratified by country and Ashkenazi Jewish ancestry, and further adjusted for genotyping array (iCOGS or OncoArray) birth cohort (<1920, 1920-29, 1930-39, 1940-49, ≥1950). To account for relatedness, cluster robust variances were estimated using unique family IDs as clusters.

**CNV validation.** Gene loci found associated with risk were reviewed to identify CNVs for validation using orthogonal technologies. CNVs were prioritised for validation if one or more DNA samples were available, the gene locus was associated with risk (unadjusted *p*-value < 0.01), and if they overlapped a gene region that had been previously associated with breast cancer risk. Copy number assessment was carried out using TaqMan assays for five different copy number variable regions in three different genes, including one gene (*SULT1A1*) that also was assayed with MLPA. Custom primer and probe sequences and pre-designed assays from Life Technologies used to validate CNVs are listed in Supplementary Data 15. *SULT1A1* MLPA was performed using the *SULT1A1* MLPA kit (C1-0217; MRC-Holland) and analysed using Coffalyser software (v9.4; MRC-Holland), as per the manufacturer's instructions.

Three samples also were evaluated using WGS to assess the genome-wide accuracy of CNV calling. Libraries were prepared for whole-genome sequencing using the KAPA Hyper PCR Free Library preparation kit (V2.1) and 2 × 150 bp paired end sequenced on the Illumina HiSeqX platform (Kinghorn Centre for Clinical Genomics, Australia). Genomic data were processed using a modified version of the GATK best practise guidelines. Cases were sequenced to an average of 30-fold depth and CNVs were called using Lumpy and CNVnator (Supplementary Data 16). Briefly, FastQ files were generated and adaptors trimmed using Illumina's Bcl2fastq (V2.16). Reads were aligned to the b37d5 (1000 Genomes Project GRCh37 plus decoy) reference genome using BWA-mem (V0.7.10-r789)[63], followed by Novosort (V1.03.01) to create coordinated-sorted duplicate marked files. GATK (V3.3) Indel realignment and base quality recalibration were used to create analysis ready reads[64]. Single nucleotide variants

and short insertions and deletions were joint-called using GATK HaplotypeCaller in gVCF mode with variant quality score recalibration. Structural variants, including CNVs, were distinguished from split reads and discordant pairs using lumpy (V0.2.13)[65] and read depth differences using CNVnator (V0.3.3)[66].

**Dosage effect analysis.** Expression and copy number data from the Breast Invasive Carcinoma[67] datasets were downloaded using cBioPortal[68]. mRNA expression was calculated as a Z-score from all genes and putative copy number alterations were calculated using GISTIC.

**Cell culture.** The MCF7 breast cancer cell line was purchased from the American Type Culture Collection (ATCC) and maintained in Dulbecco's Modified Eagle's Medium (DMEM) supplemented with 10% (v/v) foetal bovine serum (FBS). Cells were cultured in a humidified atmosphere of 5% $CO_2$ at 37 °C, and routinely passaged every 3–4 days. Cells were used up to a maximum of 30 passages.

**Development of MCF7–BRCA1$^{+/-}$ cell line.** MCF7 cells underwent CRISPR-Cas9 editing to create isogenic cells with and without a heterozygous *BRCA1* frameshift variant resulting in premature truncation of the protein (hereafter referred to as a pathogenic variant). The guide RNA was designed to target exon 11 of *BRCA1* and disrupt its function (sequence 5'-GCAGCATTTGAAAACCCCAA). MCF7 cells were transfected with plasmid containing gRNA, Cas9 protein, and puromycin resistance (Addgene [ID #62988]—pSpCas9(BB)−2A-Puro (PX459) V2.0). Control cells underwent a parallel transfection protocol with a null guide RNA plasmid. CRISPR-Cas9 treated cells were clonally expanded and the predicted CRISPR cleavage site was amplified by PCR (forward 5'- GAAAGGATCCTGGGTGTTTG, reverse 5'- CTTGTTTCCCGACTGTGGTT,) and was Sanger sequenced to identify pathogenic variants. Isogenic lines were cultured in DMEM (1:1; Invitrogen) with 10% (v/v) FBS (Invitrogen) and grown in a humidified atmosphere of 5% $CO_2$ at 37 °C.

**RNA interference.** Cells were seeded at 5000 cells per well in 96-well tissue culture plates and allowed to adhere overnight. Cells were transfected with 20 µM of Silencer™ Select siRNA oligonucleotides targeting human *SULT1A1* (s13613, Ambion) or a non-targeting siRNA negative control (Negative Control No. 1 siRNA, Ambion). Cells were transfected using Lipofectamine RNAi max (Invitrogen) according to manufacturer's specifications. After 24 h of transfection media was replaced with normal growth media.

**qPCR.** Total RNA was extracted 72 h post transfection using the RNAgem-PLUS kit (ZyGem) to assess the level of gene knockdown. cDNA was synthesized using the Superscript III cDNA Synthesis Kit (Invitrogen) and qPCR was performed using Kapa Probe Fast qPCR Master mix (Kapa Biosystems) on the LightCycler 480 (Roche). The $2^{-\Delta\Delta CT}$ method was used to quantify mRNA expression levels of target genes, where *HPRT1* was used as an internal reference control. Two well-characterised samples from 1000 Genomes Project with known copy number status were used as copy number controls. Gene-specific primers and fluorescent probes are reported in Supplementary Data 17. Statistical significance was assessed by two-tailed Student's *t*-tests between target genes and the siRNA control. Expression differences were considered statistically significant if the *p*-value was <0.05.

**Proliferation assay.** MCF7 and MCF7–BRCA1$^{+/-}$ cells were seeded at 5000 cells per well in 96-well, black walled, clear-bottom tissue culture plates (Greiner). Cells were allowed to adhere overnight before transfection. Media was replaced 24 h post transfection. Forty-eight hours post media change, cell proliferation was assessed using the CyQUANT™ Cell Proliferation Assay Kit (Invitrogen) according to the manufacturer's instructions. Fluorescence was measured on the Varioskan® Flash plate reader (Thermo Fisher Scientific) using a filter combination for excitation at 480 nm and emission at 520 nm.

**DNA damage assay.** Cells were seeded at 50,000 cells per well in 24-well tissue culture plates. Cells were allowed to adhere overnight and were transfected for 24 h before media was replaced. A further 24 h after media replacement cells were treated with 1 µM 4-hydroxyestradiol (4-OHE2, Sigma) or 10 µM Mitomycin C (MMC, Sigma) for 3 h. Cells were gently washed with PBS and media was replaced with fresh complete media for a further 21 h before being assayed for DNA damage.

**Immunocytochemistry.** Cells were gently lifted from cell culture plates, cytospun onto slides, and fixed in ice-cold absolute methanol for 5 min. Slides were washed with PBS and blocked for 30 min with 1% bovine serum albumin in PBS-T (Tween-20 0.1% v/v). Slides were dual stained for 1 h with the mouse anti-phospho-H2AX (Ser139) antibody (1:500; ab26350, Abcam) and rabbit anti-53BP1 (1:500; ab36823, Abcam). Slides were incubated with anti-mouse AlexaFluor 488-conjugated (1:400; ab150113, Abcam) and anti-rabbit IgG-AlexaFluor 494-conjugated (1:400; ab150080, Abcam) secondary antibodies, and stained with DAPI for microscopic

examination. Images were taken at 40 × magnification on the Axio Imager.Z1 Microscope (Zeiss). Co-localised ϒ-H2AX and 53BP1 foci were counted in >150 cells from a minimum of ten fields from three independent experiments.

**Comet assay.** Alkaline comet assays were performed using a comet assay kit (AbCam). Harvested cells were mixed with low melting agarose and transferred onto a glass slide covered in a base layer of agarose. Slides were immersed in lysis buffer for 60 min at 4 °C. Lysis buffer was replaced with alkaline solution (300 mM NaOH, pH 10, 1 mM EDTA) and samples were kept in the dark for 30 min. Slides were transferred to an electrophoresis chamber filled with alkaline solution and electrophoresis was performed for 20 min (1 V/cm). DNA was stained with Vista Green DNA Dye and images were captured by fluorescence microscopy on the Axio Imager.Z1 Microscope. Comets were scored using the CellProfiler software v3.1.8[69]. Tail moments were assessed for >100 cells in three independent experiments.

**Statistical analysis of in vitro data.** All in vitro data were expressed as the mean ± standard error. The normality of data was visualised using the Q–Q plot and tested using the Shapiro–Wilk normality test. Statistical significance of differences between control and test groups were determined by an unpaired Student's *t*-test or two-way analysis of variance (ANOVA). All statistical tests were two sided and *p*-values < 0.05 were considered significant.

**Reporting summary.** Further information on research design is available in the Nature Research Reporting Summary linked to this article.

## Data availability
Genome-wide association summary statistics are available within the article. CIMBA phenotype data used in this study from BCFR-AU, BCFR-NC, BCFR-NY, BCFR-PA, BCFR-UT, BFBOCC, BIDMC, BMBSA, CBCS, CNIO, COH, DEMOKRITOS, DFCI, FCCC, GEORGETOWN, HCSC, HRBCP, HUNBOCS, HVH, ICO, KCONFAB, KUMC, MAYO, MSKCC, MUV, NCI, NNPIO, NORTHSHORE, OSUCCG, PBCS, SMC, SWE-BRCA, UCHICAGO, UCSF, UPENN, UPITT, UTMDACC, VFCTG, and WCP studies are available in the dbGaP database under accession code phs001321.v1.p1. The complete dataset is not publicly available due to restraints imposed by the ethical committees of individual studies. Requests to access the complete dataset, which is subject to General Data Protection Regulation (GDPR) rules, can be made to the Data Access Coordinating Committee (DACC) of CIMBA, following the process described on the CIMBA website (http://cimba.ccge.medschl.cam.ac.uk/projects/data-access-requests/). Submitted applications are reviewed by the CIMBA DACC every 3 months. CIMBA DACC approval is required to access data from studies BCFR-ON/OCGN, BFBOCC-LV, BRICOH, CCGCRN, BRICOH, CONSIT TEAM, DKFZ, EMBRACE, FPGMX, GC-HBOC, GEMO, G-FAST, HEBCS, HEBON, IHCC, ILUH, INHERIT, IOVHBOCS, IPOBCS, KOHBRA, MCGILL, NCCS, NRG_ONCOLOGY, OUH, SEABASS, and UKGRFOCR (see Supplementary Data 12 —for a list of all CIMBA studies). Summary statistics for each GWAS conducted for this study, can be freely downloaded from the NHGRI-EBI GWAS catalogue with the accession codes: GCST90134567; GCST90134568; GCST90134569; and GCST90134570; (https://www.ebi.ac.uk/gwas/). The source data for all figures are presented in the Supplementary Data file.

## Code availability
Code for the retrospective likelihood analysis of variants is available on request.

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

## Acknowledgements

All the families and clinicians who contribute to the studies; Catherine M. Phelan for her contribution to CIMBA until she passed away on 22 September 2017; Sue Healey, in particular taking on the task of mutation classification with the late Olga Sinilnikova; Maggie Angelakos, Judi Maskiell, Gillian Dite, Helen Tsimiklis; members and participants in the New York site of the Breast Cancer Family Registry; members and participants in the Ontario Familial Breast Cancer Registry; Vilius Rudaitis and Laimonas Griškevičius; Drs Janis Eglitis, Anna Krilova and Aivars Stengrevics; Yuan Chun Ding and Linda Steele for their work in participant enrolment and biospecimen and data management; Bent Ejlertsen and Anne-Marie Gerdes for the recruitment and genetic counselling of participants; Alicia Barroso, Rosario Alonso and Guillermo Pita; all the individuals and the researchers who took part in CONSIT TEAM (Consorzio Italiano Tumori Ereditari Alla Mammella), in particular: Bernard Peissel, Dario Zimbalatti, Daniela Zaffaroni, Laura Ottini, Giuseppe Giannini and the personnel of the Cogentech Cancer Genetic Test Laboratory, Milan, Italy. Ms. JoEllen Weaver and Dr. Betsy Bove; FPGMX: members of the Cancer Genetics group (IDIS): Miguel Aguado, Olivia Fuentes, and Ana Crujeiras; IFE—Leipzig Research Centre for Civilization Diseases (Markus Loeffler, Joachim Thiery, Matthias Nüchter, Ronny Baber); We thank all participants, clinicians, family doctors, researchers, and technicians for their contributions and commitment to the DKFZ study and the collaborating groups in Lahore, Pakistan (Muhammad U. Rashid, Noor Muhammad, Sidra Gull, Seerat Bajwa, Faiz Ali Khan, Humaira Naeemi, Saima Faisal, Asif Loya, Mohammed Aasim Yusuf) and Bogota, Colombia (Diana Torres, Ignacio Briceno, Fabian Gil). Genetic Modifiers of Cancer Risk in BRCA1/2 Mutation Carriers (GEMO) study is a study from the National Cancer Genetics Network UNICANCER Genetic Group, France. We wish to pay a tribute to Olga M. Sinilnikova, who with Dominique Stoppa-Lyonnet initiated and coordinated GEMO until she sadly passed away on the 30th June 2014. The team in Lyon (Olga Sinilnikova, Mélanie Léoné, Laure Barjhoux, Carole Verny-Pierre, Sylvie Mazoyer, Francesca Damiola, Valérie Sornin) managed the GEMO samples until the biological resource centre was transferred to Paris in December 2015 (Noura Mebirouk, Fabienne Lesueur, Dominique Stoppa-Lyonnet). We want to thank all the GEMO collaborating groups for their contribution to this study: Coordinating Centre, Service de Génétique, Institut Curie, Paris, France: Muriel Belotti, Ophélie Bertrand, Anne-Marie Birot, Bruno Buecher, Sandrine Caputo, Chrystelle Colas, Anaïs Dupré, Emmanuelle Fourme, Marion Gauthier-Villars, Lisa Golmard, Marine Le Mentec, Virginie Moncoutier, Antoine de Pauw, Claire Saule, Dominique Stoppa-Lyonnet, and Inserm U900, Institut Curie, Paris, France: Fabienne Lesueur, Noura Mebirouk.Contributing Centres: Unité Mixte de Génétique Constitutionnelle des Cancers Fréquents, Hospices Civils de Lyon—Centre Léon Bérard, Lyon, France: Nadia Boutry-Kryza, Alain Calender, Sophie Giraud, Mélanie Léone. Institut Gustave Roussy, Villejuif, France: Brigitte Bressac-de-Paillerets, Olivier Caron, Marine Guillaud-Bataille. Centre Jean Perrin, Clermont–Ferrand, France: Yves-Jean Bignon, Nancy Uhrhammer. Centre Léon Bérard, Lyon, France: Valérie Bonadona, Christine Lasset. Centre François Baclesse, Caen, France: Pascaline Berthet, Laurent Castera, Dominique Vaur. Institut Paoli Calmettes, Marseille, France: Violaine Bourdon, Catherine Noguès, Tetsuro Noguchi, Cornel Popovici, Audrey Remenieras, Hagay Sobol. CHU Arnaud-de-Villeneuve, Montpellier, France: Isabelle Coupier, Pascal Pujol. Centre Oscar Lambret, Lille, France: Claude Adenis, Aurélie Dumont, Françoise Révillion. Centre Paul Strauss, Strasbourg, France: Danièle Muller. Institut Bergonié, Bordeaux, France: Emmanuelle Barouk-Simonet, Françoise Bonnet, Virginie Bubien, Michel Longy, Nicolas Sevenet, Institut Claudius Regaud, Toulouse, France: Laurence Gladieff, Rosine Guimbaud, Viviane Feillel, Christine Toulas. CHU Grenoble, France: Hélène Dreyfus, Christine Dominique Leroux, Magalie Peysselon, Rebischung. CHU Dijon, France: Amandine Baurand, Geoffrey Bertolone, Fanny Coron, Laurence Faivre, Caroline Jacquot, Sarab Lizard. CHU St-Etienne, France: Caroline Kientz, Marine Lebrun, Fabienne Prieur. Hôtel Dieu Centre Hospitalier, Chambéry, France: Sandra Fert Ferrer. Centre Antoine Lacassagne, Nice, France: Véronique Mari. CHU Limoges, France: Laurence Vénat-Bouvet. CHU Nantes, France: Stéphane Bézieau, Capucine Delnatte. CHU Bretonneau, Tours and Centre Hospitalier de Bourges France: Isabelle Mortemousque. Groupe Hospitalier Pitié-Salpêtrière, Paris, France: Florence Coulet, Mathilde Warcoin. CHU Vandoeuvre-les-Nancy, France: Myriam Bronner, Johanna Sokolowska. CHU Besançon, France: Marie-Agnès Collonge-Rame, Alexandre Damette. CHU Poitiers, Centre Hospitalier d'Angoulême and Centre Hospitalier de Niort, France: Paul Gesta. Centre Hospitalier de La Rochelle, France: Hakima Lallaoui. CHU Nîmes Carémeau, France: Jean Chiesa. CHI Poissy, France: Denise Molina-Gomes. CHU Angers, France: Olivier Ingster; CHU de Martinique, France: Odile Bera; Mickaelle Rose; Ilse Coene and Brecht Crombez; Alicia Tosar and Paula Diaque; Virpi Palola; The Hereditary Breast and Ovarian Cancer Research Group Netherlands (HEBON) consists of the following Collaborating Centres: Netherlands Cancer Institute (coordinating centre), Amsterdam, NL: M.A. Rookus, F.B.L. Hogervorst, F.E. van Leeuwen, M.A. Adank, M.K. Schmidt, D.J. Jenner; Erasmus Medical Center, Rotterdam, NL: J.M. Collée, A.M.W. van den Ouweland, M.J. Hooning, I.A. Boere; Leiden University Medical Center, NL: C.J. van Asperen, P. Devilee, R.B. van der Luijt, T.C.T.E.F. van Cronenburg; Radboud University Nijmegen Medical Center, NL: M.R. Wevers, A.R. Mensenkamp; University Medical Center Utrecht, NL: M.G.E.M. Ausems, M.J. Koudijs; Amsterdam UMC, Univ of Amsterdam, NL: I. van de Beek; Amsterdam UMC, Vrije Universiteit Amsterdam, NL: K. van Engelen, J.J.P. Gille; Maastricht University Medical Center, NL: E.B. Gómez García, M.J. Blok, M. de Boer; University of Groningen, NL: L.P.V. Berger, A.H. van der Hout, M.J.E. Mourits, G.H. de Bock; The Netherlands Comprehensive Cancer Organisation (IKNL): S. Siesling, J. Verloop; The nationwide network and registry of histo- and cytopathology in The Netherlands (PALGA): E.C. van den Broek. HEBCS thanks Drs. Taru A. Muranen and Carl Blomqvist and research nurse Irja Erkkilä. HEBON thanks the study participants and the registration teams of IKNL and PALGA for part of the data collection; Hong Kong Sanatorium and Hospital; the Hungarian Breast and Ovarian Cancer Study Group members (Janos Papp, Aniko Bozsik, Department of Molecular Genetics, National Institute of Oncology, Budapest, Hungary) and the clinicians and patients for their contributions to this study; Hereditary Cancer Genetics Group, Vall d'Hebron Institute of Oncology (VHIO), the High Risk and Cancer Prevention Unit of the University Hospital Vall d'Hebron, and the Cellex Foundation for providing research facilities and equipment; the ICO Hereditary Cancer Program team led by Dr. Gabriel Capella; Dr Martine Dumont for sample management and skilful assistance; Catarina Santos and Pedro Pinto; members of the Center of Molecular Diagnosis, Oncogenetics Department and Molecular Oncology Research Center of Barretos Cancer Hospital; Heather Thorne, Eveline Niedermayr, all the kConFab research nurses and staff, the heads and staff of the Family Cancer Clinics, and the Clinical Follow Up Study (which has received funding from the NHMRC, the National Breast Cancer Foundation, Cancer Australia, and the National Institute of Health (USA)) for their contributions to this resource, and the many families who contribute to kConFab; the KOBRA Study Group; Csilla Szabo (National Human Genome Research Institute, National Institutes of Health, Bethesda, MD, USA); Lenka Foretova and Eva Machackova (Department of Cancer Epidemiology and Genetics, Masaryk Memorial Cancer Institute and MF MU, Brno, Czech Republic); and Michal Zikan, Petr Pohlreich and Zdenek Kleibl (Oncogynecologic Center and Department of Biochemistry and Experimental Oncology, First Faculty of Medicine, Charles University, Prague, Czech Republic); Anne Lincoln, Lauren Jacobs; the participants in Hereditary Breast/Ovarian Cancer Study and Breast Imaging Study for their selfless contributions to our research; the NICCC National Familial Cancer Consultation Service team led by Sara Dishon, the lab team led by Dr. Flavio Lejbkowicz, and the research field operations team led by Dr. Mila Pinchev; the investigators of the Australia New Zealand NRG Oncology group; members and participants in the Ontario Cancer Genetics Network; Leigha Senter, Kevin Sweet, Caroline Craven, Julia Cooper, Amber Aielts, and Michelle O'Conor; Yip Cheng Har, Nur Aishah Mohd Taib, Phuah Sze Yee, Norhashimah Hassan and all the research nurses, research assistants and doctors involved in the MyBrCa Study for assistance in patient recruitment, data collection and sample preparation, Philip Iau, Sng Jen-Hwei and Sharifah Nor Akmal for contributing samples from the Singapore Breast Cancer Study and the HUKM-HKL Study respectively; the Meirav Comprehensive breast cancer center team at the Sheba Medical Center; Christina Selkirk; Håkan Olsson, Helena Jernström, Karin Henriksson, Katja Harbst, Maria Soller, Ulf Kristofferson; from Gothenburg Sahlgrenska University Hospital: Anna Öfverholm, Margareta Nordling, Per Karlsson, Zakaria Einbeigi; from Stockholm and Karolinska University Hospital: Anna von Wachenfeldt, Annelie Liljegren, Annika Lindblom, Brita Arver, Gisela Barbany Bustinza from Umeå University Hospital: Beatrice Melin, Christina Edwinsdotter Ardnor, Monica Emanuelsson; from Uppsala University: Hans Ehrencrona, Maritta Hellström Pigg, Richard Rosenquist; from Linköping University Hospital: Marie Stenmark-Askmalm, Sigrun Liedgren; Cecilia Zvocec, Qun Niu; Joyce Seldon and Lorna Kwan; Dr. Robert Nussbaum, Beth Crawford, Kate Loranger, Julie Mak, Nicola Stewart, Robin Lee, Amie Blanco and Peggy Conrad and Salina Chan; Paul Pharoah, Carole Pye, Patricia Harrington and Eva Wozniak; Geoffrey Lindeman, Marion Harris, Martin Delatycki, Sarah Sawyer, Rebecca Driessen, and Ella Thompson for performing all DNA amplification. This study was supported by a grant from the New

Zealand Health Research Council (#17/113) and the Mackenzie Charitable Foundation. CIMBA: The CIMBA data management and data analysis were supported by Cancer Research—UK grants C12292/A20861 and PPRPGM-Nov20\100002 and the Gray Foundation. GCT and ABS are NHMRC Research Fellows. iCOGS: the European Community's Seventh Framework Programme under grant agreement n° 223175 (HEALTH-F2-2009-223175) (COGS), Cancer Research UK (C1287/A10118, C1287/A 10710, C12292/A11174, C1281/A12014, C5047/A8384, C5047/A15007, C5047/A10692, C8197/A16565), the National Institutes of Health (CA128978) and Post-Cancer GWAS initiative (1U19 CA148537, 1U19 CA148065 and 1U19 CA148112—the GAME-ON initiative), the Department of Defence (W81XWH-10-1-0341), the Canadian Institutes of Health Research (CIHR) for the CIHR Team in Familial Risks of Breast Cancer (CRN-87521), and the Ministry of Economic Development, Innovation and Export Trade (PSR-SIIRI-701), Komen Foundation for the Cure, the Breast Cancer Research Foundation, and the Ovarian Cancer Research Fund. OncoArray: the PERSPECTIVE and PER-SPECTIVE I&I projects funded by the Government of Canada through Genome Canada and the Canadian Institutes of Health Research, the 'Ministère de l'Économie, de la Science et de l'Innovation du Québec' through Genome Québec, and the Quebec Breast Cancer Foundation; the NCI Genetic Associations and Mechanisms in Oncology (GAME-ON) initiative and Discovery, Biology and Risk of Inherited Variants in Breast Cancer (DRIVE) project (NIH Grants U19 CA148065 and X01HG007492); and Cancer Research UK (C1287/A10118 and C1287/A16563). BCFR: UM1 CA164920 from the National Cancer Institute. The content of this manuscript does not necessarily reflect the views or policies of the National Cancer Institute or any of the collaborating centres in the Breast Cancer Family Registry (BCFR), nor does mention of trade names, commercial products, or organizations imply endorsement by the US Government or the BCFR. BFBOCC: Lithuania (BFBOCC-LT): Research Council of Lithuania grant SEN-18/2015. BIDMC: Breast Cancer Research Foundation. BMBSA: Cancer Association of South Africa (PI Elizabeth J. van Rensburg). CNIO study is partially funded by FIS PI19/00640 supported by FEDER funds, and the Spanish Network on Rare Diseases (CIBERER) CCGCRN: Research reported in this publication was supported by the Breast Cancer Research Foundation (project 20-172), National Cancer Institute of the National Institutes of Health under grant number R25CA112486, and RC4CA153828 (PI: J. Weitzel) from the National Cancer Institute and the Office of the Director, National Institutes of Health. The content is solely the responsibility of the authors and does not necessarily represent the official views of the National Institutes of Health. CONSIT TEAM: Associazione Italiana Ricerca sul Cancro (AIRC; IG2015 no.16732) to P. Peterlongo and P. Radice. DEMOKRITOS: European Union (European Social Fund—ESF) and Greek national funds through the Operational Program "Education and Life-long Learning" of the National Strategic Reference Framework (NSRF)—Research Funding Program of the General Secretariat for Research & Technology: SYN11_10_19 NBCA. Investing in knowledge society through the European Social Fund. DFKZ: German Cancer Research Center. EMBRACE: Cancer Research UK Grants C1287/A23382 and C1287/A26886. D. Gareth Evans and Fiona Lalloo are supported by an NIHR grant to the Biomedical Research Centre, Manchester (IS-BRC-1215-20007). The Investigators at The Institute of Cancer Research and The Royal Marsden NHS Foundation Trust are supported by an NIHR grant to the Biomedical Research Centre at The Institute of Cancer Research and The Royal Marsden NHS Foundation Trust. Ros Eeles and Elizabeth Bancroft are supported by Cancer Research UK Grant C5047/A8385. Ros Eeles is also supported by NIHR support to the Biomedical Research Centre at The Institute of Cancer Research and The Royal Marsden NHS Foundation Trust. FCCC: The University of Kansas Cancer Center (P30 CA168524) the Kansas Institute for Precision Medicine (P20GM130423) and the Kansas Bioscience Authority Eminent Scholar Program. A.K.G. was funded by R01CA140323, R01CA214545, R01CA260132, 5U10CA180888, and by the Chancellors Distinguished Chair in Biomedical Sciences Professorship. FPGMX: A.Vega is supported by Spanish Instituto de Salud Carlos III (ISCIII) funding, an initiative of the Spanish Ministry of Economy and Innovation partially supported by European Regional Development FEDER Funds (INT20/00071, INT17/00133, INT16/00154, INT15/00070; PI19/01424; PI16/00046; PI13/02030; PI10/00164), through the Autonomous Government of Galicia (Consolidation and structuring program: IN607B), through Centro de Investigación Biomédica en Red de Enferemdades Raras CIBERER (ACCI 2016: ER17P1AC7112/2018) and by the Fundación Mutua Madrileña (call 2018). GC-HBOC: German Cancer Aid (grant no 110837 and 113049, Rita K. Schmutzler) and the European Regional Development Fund and Free State of Saxony, Germany (LIFE—Leipzig Research Centre for Civilization Diseases, project numbers 713-241202, 713-241202, 14505/2470, 14575/2470). GEMO: Ligue Nationale Contre le Cancer; the Association "Le cancer du sein, parlons-en!" Award, the Canadian Institutes of Health Research for the "CIHR Team in Familial Risks of Breast Cancer" program, the Fondation ARC pour la recherche sur le cancer (grant PJA 20151203365) and the French National Institute of Cancer (INCa grants AOR 01 082, 2001-2003, 2013-1-BCB-01-ICH-1 and SHS-E-SP 18-015). GEORGETOWN: the Survey, Recruitment, and Biospecimen Shared Resource at Georgetown University (NIH/NCI grant P30-CA051008) and the Fisher Center for Hereditary Cancer and Clinical Genomics Research. G-FAST: Bruce Poppe is a senior clinical investigator of FWO. Mattias Van Heetvelde obtained funding from IWT. HCSC: Spanish Ministry of Health PI15/00059, PI16/01292, and CB-161200301 CIBERONC from ISCIII (Spain), partially supported by European Regional Development FEDER funds. HEBCS: Helsinki University Hospital Research Fund, the Finnish Cancer Society and the Sigrid Juselius Foundation. HEBON: the Dutch Cancer Society grants NKI1998-1854, NKI2004-3088, NKI2007-3756, the

Netherlands Organization of Scientific Research grant NWO 91109024, the Pink Ribbon grants 110005 and 2014-187.WO76, the BBMRI grant NWO 184.021.007/CP46 and the Transcan grant JTC 2012 Cancer 12-054. HEBON thanks the registration teams of Dutch Cancer Registry (IKNL; S. Siesling, J. Verloop) and the Dutch Pathology database (PALGA; L. Overbeek) for part of the data collection. HRBCP: Hong Kong Sanatorium and Hospital, Dr Ellen Li Charitable Foundation, The Kerry Group Kuok Foundation, National Institute of Health1R 03CA130065, and North California Cancer Center. HUNBOCS: Hungarian Research Grants KTIA-OTKA CK-80745 and NKFI_OTKA K-112228. (E. Olah) and 2019 Thematic Excellence Program (TUDFO/51757/2019-ITM) ICO: Supported by the Carlos III National Health Institute and Ministerio de Ciencia e Innovación, funded by FEDER funds—a way to build Europe—[PI19/00553; PI16/00563; PI18/01029 and CIBERONC]; the Government of Catalonia [Pla estratègic de recerca i innovació en salut (PERIS_MedPerCan and URDCat projects), 2017SGR1282 and 2017SGR496]. IHCC: The IHCC study was supported by Grant PBZ_KBN_122/P05/2004 and the program of the Minister of Science and Higher Education under the name "Regional Initiative of Excellence" in 2019–2022 project number 002/RID/2018/19 amount of financing 12,000,000 PLN. ILUH: Icelandic Association "Walking for Breast Cancer Research" and by the Landspitali University Hospital Research Fund. INHERIT: Canadian Institutes of Health Research for the "CIHR Team in Familial Risks of Breast Cancer" program—grant # CRN-87521 and the Ministry of Economic Development, Innovation and Export Trade—grant # PSR-SIIRI-701. IOVHBOCS: Ministero della Salute and "5 × 1000" Istituto Oncologico Veneto grant. IPOBCS: Liga Portuguesa Contra o Cancro. kConFab: The National Breast Cancer Foundation, and previously by the National Health and Medical Research Council (NHMRC), the Queensland Cancer Fund, the Cancer Councils of New South Wales, Victoria, Tasmania and South Australia, and the Cancer Foundation of Western Australia. KOHBRA: the Korea Health Technology R&D Project through the Korea Health Industry Development Institute (KHIDI), and the National R&D Program for Cancer Control, Ministry of Health & Welfare, Republic of Korea (HI16C1127; 1020350; 1420190). KUMC: NIGMS P20 GM130423 (to AKG). MAYO: NIH grants CA116167, CA192393 and CA176785, an NCI Specialized Program of Research Excellence (SPORE) in Breast Cancer (CA116201),and a grant from the Breast Cancer Research Foundation. MCGILL: Jewish General Hospital Weekend to End Breast Cancer, Quebec Ministry of Economic Development, Innovation and Export Trade. Marc Tischkowitz is supported by the funded by the European Union Seventh Framework Program (2007Y2013)/European Research Council (Grant No. 310018). MODSQUAD: MH CZ—DRO (MMCI, 00209805), MEYS—NPS I—LO1413 to LF, and by Charles University in Prague project UNCE204024 (MZ). MSKCC: the Breast Cancer Research Foundation, the Robert and Kate Niehaus Clinical Cancer Genetics Initiative, the Andrew Sabin Research Fund and a Cancer Center Support Grant/Core Grant (P30 CA008748). NAROD: 1R01 CA149429-01. NCI: the Intramural Research Program of the US National Cancer Institute, NIH, supports Mark H Greene and Megan Frone, and by support services contracts NO2-CP-11019-50, N02-CP-21013-63 and N02-CP-65504 with Westat, Inc, Rockville, MD. NICCC: Clalit Health Services in Israel, the Israel Cancer Association and the Breast Cancer Research Foundation (BCRF), NY. NNPIO: the Russian Science Foundation (grant number 21-75-30015). NRG Oncology: U10 CA180868, NRG SDMC grant U10 CA180822, NRG Administrative Office and the NRG Tissue Bank (CA 27469), the NRG Statistical and Data Center (CA 37517) and the Intramural Research Program, NCI. OSUCCG: Ohio State University Comprehensive Cancer Center. PBCS: Italian Association of Cancer Research (AIRC) [IG 2013 N.14477] and Tuscany Institute for Tumors (ITT) grant 2014-2015-2016. SEABASS: Ministry of Science, Technology and Innovation, Ministry of Higher Education (UM.C/HlR/MOHE/06) and Cancer Research Initiatives Foundation. SMC: the Israeli Cancer Association. SWE-BRCA: the Swedish Cancer Society. UCHICAGO: NCI Specialized Program of Research Excellence (SPORE) in Breast Cancer (CA125183), R01 CA142996, 1U01CA161032 and by the Ralph and Marion Falk Medical Research Trust, the Entertainment Industry Fund National Women's Cancer Research Alliance and the Breast Cancer research Foundation. O.I.O. is an ACS Clinical Research Professor. UCLA: Jonsson Comprehensive Cancer Center Foundation; Breast Cancer Research Foundation. UCSF: UCSF Cancer Risk Program and Helen Diller Family Comprehensive Cancer Center. UKFOCR: Cancer Researc h UK. UPENN: Breast Cancer Research Foundation; Susan G. Komen Foundation for the cure, Basser Research Center for BRCA. UPITT/MWH: Hackers for Hope Pittsburgh. VFCTG: Victorian Cancer Agency, Cancer Australia, National Breast Cancer Foundation. WCP: Dr Karlan is funded by the American Cancer Society Early Detection Professorship (SIOP-06-258-01-COUN) and the National Center for Advancing Translational Sciences (NCATS), Grant UL1TR000124. L.C.W. was funded by the Rutherford Discovery Fellowship (Royal Society of New Zealand)—RDF-UOO1501. A.B.S. was supported by an NHMRC Investigator Fellowship (APP177524). T.N.-D. is a National Breast Cancer Foundation (Australia) Career Development Fellow. J.D. is supported by the CanRisk Cancer Research UK programme grant: PPRPGM-Nov20\100002 and by the Confluence project which is funded with intramural funds from the National Cancer Institute Intramural Research Program, National Institutes of Health. MT was supported by the NIHR Cambridge Biomedical Research Centre (BRC-1215-20014).

## Author contributions

L.C.W., A.C.A., A.B.S., J.F.P., B.A.R., and P.D.M. conceived the study design. C.H. and L.C.W. drafted the initial manuscript, while the complete writing group consisted of

C.H., L.C.W., A.C.A, A.B.S., L.Ma., J.D. G.C.-T., J.F.P., B.A.R., P.D.M., and D.R.B. L.Ma. and D.R.B performed the statistical analyses. J.D. conducted the CNV calling. C.H. conducted the laboratory experiments and G.A.R.W. generated the genetically modified cell line. J.D., L.Mc., and G.Le. performed the data management. K.Ai., I.L.A., B.K.A., J.A., J.Ba., R.B.B., S.B., L.B., M.J.B., S.E.B., J.Bo., A.R.B., J.Br., S.S.B., M.A.C., I.C., W.K.C., K.B.M.C., GEMO Study Collaborators, EMBRACE Collaborators, M.-A.C.-R., J.Co., C.C., F.J.C., M.B.D., S.D., R.Da., M.d.l.H., R.d.P., C.D., M.D., O.D., Y.C.D., S.M.D., A.Do., J.E., D.F.E., H.E., C.E., D.G.E., U.F., L.F., F.F., E.Fr., M.F., D.F., J.G., S.A.G., A.G., P.Ge., A.K.G., D.E.G., M.H.G., E.Ha., C.H., U.H., T.V.O.H., J.Ha., J.He., N.He., E.Hon., P.J.H., E.N.I., SWE-BRCA Investigators, kConFab Investigators, HEBON Investigators, C.I., L.I., A.I., A.Ja., P.A.J., R.J., E.M.J., V.J., B.Y.K., Z.K., J.K., I.K., M.K., A.K., Y.L., F.L., C.Las., C.Lau., C.Laz., C.Le, F.Les., P.L.M., S.M., V.M., J.W.M.M., N.M., A.Me., A.Mil., M.M., L.Mo., E.M.-F., H.M., S.N., K.L.N., S.L.N., H.N., J.N.Y.Y., T.N.-D., L.N.-Z., K.O., E.O., O.L.O., A.O., C.-E.O., S.K.P., M.T.P., I.S.P., A.P., P.P.-S., P.P., T.P., P.R., J.Ram., J.Ran., G.C.R., K.R., E.H.R., M.R., R.K.S., P.D.S., S.S., P.S., L.E.S., J.Si., C.F.S., K.S., D.S., D.S.-L., C.S., Y.Y.T., M.R.T., S.H.T., M.Th., D.L.T., M.Ti., A.E.T., A.H.T., V.T., N.T., K.v.E., E.J.v.R., A.Ve., A.Vi., L.W., J.N.W., and M.R.W provided DNA samples and phenotypic data. All authors read and approved the final manuscript. The funders had no role in the design of the study, the collection, analysis, or interpretation of the data, the writing of the manuscript, or the decision to submit the manuscript for publication.

## Competing interests

The authors declare no competing interests

## Additional information

Christopher Hakkaart[1], John F. Pearson[1], Louise Marquart[2,3], Joe Dennis[4], George A. R. Wiggins[1], Daniel R. Barnes[4], Bridget A. Robinson[5,6], Peter D. Mace[7], Kristiina Aittomäki[8], Irene L. Andrulis[9,10], Banu K. Arun[11], Jacopo Azzollini[12], Judith Balmaña[13,14], Rosa B. Barkardottir[15,16], Sami Belhadj[17], Lieke Berger[18], Marinus J. Blok[19], Susanne E. Boonen[20], Julika Borde[21,22,23], Angela R. Bradbury[24], Joan Brunet[25], Saundra S. Buys[26], Maria A. Caligo[27], Ian Campbell[28,29], Wendy K. Chung[30], Kathleen B. M. Claes[31], GEMO Study Collaborators*, EMBRACE Collaborators*, Marie-Agnès Collonge-Rame[32], Jackie Cook[33], Casey Cosgrove[34], Fergus J. Couch[35], Mary B. Daly[36], Sita Dandiker[17], Rosemarie Davidson[37], Miguel de la Hoya[38], Robin de Putter[31], Capucine Delnatte[39], Mallika Dhawan[40], Orland Diez[13,41], Yuan Chun Ding[42], Susan M. Domchek[43], Alan Donaldson[44], Jacqueline Eason[45], Douglas F. Easton[4,46], Hans Ehrencrona[47,48], Christoph Engel[49,50], D. Gareth Evans[51,52], Ulrike Faust[53], Lidia Feliubadaló[25], Florentia Fostira[54], Eitan Friedman[55,56], Megan Frone[57], Debra Frost[4], Judy Garber[58], Simon A. Gayther[59], Andrea Gehrig[60], Paul Gesta[61], Andrew K. Godwin[62], David E. Goldgar[63], Mark H. Greene[57], Eric Hahnen[21,23], Christopher R. Hake[64], Ute Hamann[65], Thomas V. O. Hansen[66], Jan Hauke[21,22,23], Julia Hentschel[67], Natalie Herold[21,22,23], Ellen Honisch[68], Peter J. Hulick[69,70], Evgeny N. Imyanitov[71], SWE-BRCA Investigators*, kConFab Investigators*, HEBON Investigators*, Claudine Isaacs[72], Louise Izatt[73], Angel Izquierdo[25], Anna Jakubowska[74,75], Paul A. James[29,76], Ramunas Janavicius[77,78], Esther M. John[79,80], Vijai Joseph[17], Beth Y. Karlan[81], Zoe Kemp[82], Judy Kirk[83], Irene Konstantopoulou[54], Marco Koudijs[84], Ava Kwong[85,86,87], Yael Laitman[55], Fiona Lalloo[88], Christine Lasset[89], Charlotte Lautrup[90], Conxi Lazaro[25], Clémentine Legrand[91], Goska Leslie[4], Fabienne Lesueur[92,93,94], Phuong L. Mai[95], Siranoush Manoukian[12], Véronique Mari[96], John W. M. Martens[97], Lesley McGuffog[4], Noura Mebirouk[92,93,94], Alfons Meindl[98], Austin Miller[99], Marco Montagna[100], Lidia Moserle[100], Emmanuelle Mouret-Fourme[101], Hannah Musgrave[102], Sophie Nambot[103], Katherine L. Nathanson[43], Susan L. Neuhausen[42], Heli Nevanlinna[104], Joanne Ngeow Yuen Yie[105,106], Tu Nguyen-Dumont[107,108], Liene Nikitina-Zake[109], Kenneth Offit[17,110], Edith Olah[111], Olufunmilayo I. Olopade[112], Ana Osorio[113], Claus-Eric Ott[114], Sue K. Park[115,116,117],

Michael T. Parsons[118], Inge Sokilde Pedersen[119,120,121], Ana Peixoto[122], Pedro Perez-Segura[38], Paolo Peterlongo[123], Timea Pocza[111], Paolo Radice[124], Juliane Ramser[125], Johanna Rantala[126], Gustavo C. Rodriguez[127], Karina Rønlund[128], Efraim H. Rosenberg[129], Maria Rossing[130,131], Rita K. Schmutzler[21,22,23], Payal D. Shah[24], Saba Sharif[132], Priyanka Sharma[133], Lucy E. Side[134], Jacques Simard[135], Christian F. Singer[136], Katie Snape[137], Doris Steinemann[138], Dominique Stoppa-Lyonnet[101,139,140], Christian Sutter[141], Yen Yen Tan[136], Manuel R. Teixeira[122,142], Soo Hwang Teo[143,144], Mads Thomassen[20], Darcy L. Thull[145], Marc Tischkowitz[146,147], Amanda E. Toland[148], Alison H. Trainer[76,149], Vishakha Tripathi[150], Nadine Tung[151], Klaartje van Engelen[152], Elizabeth J. van Rensburg[153], Ana Vega[154,155,156], Alessandra Viel[157], Lisa Walker[158], Jeffrey N. Weitzel[159], Marike R. Wevers[160], Georgia Chenevix-Trench[118], Amanda B. Spurdle[118], Antonis C. Antoniou[4] & Logan C. Walker[1✉]

[1]Department of Pathology and Biomedical Science, University of Otago, Christchurch, New Zealand. [2]QIMR Berghofer Medical Research Institute, Brisbane, Queensland, Australia. [3]School of Public Health, University of Queensland, Brisbane, Australia. [4]Centre for Cancer Genetic Epidemiology, Department of Public Health and Primary Care, University of Cambridge, Cambridge, UK. [5]Department of Medicine, University of Otago, Christchurch, New Zealand. [6]Canterbury Regional Cancer and Haematology Service, Canterbury District Health Board, Christchurch Hospital, Christchurch, New Zealand. [7]Department of Biochemistry, School of Biomedical Sciences, University of Otago, Dunedin, New Zealand. [8]Department of Medical and Clinical Genetics, University of Helsinki, Helsinki, Finland. [9]Fred A. Litwin Center for Cancer Genetics, Lunenfeld-Tanenbaum Research Institute of Mount Sinai Hospital, Toronto, ON, Canada. [10]Department of Molecular Genetics, University of Toronto, Toronto, ON, Canada. [11]Department of Breast Medical Oncology, University of Texas MD Anderson Cancer Center, Houston, TX, USA. [12]Unit of Medical Genetics, Department of Medical Oncology and Hematology,  Fondazione IRCCS Istituto Nazionale dei Tumori (INT), Milan, Italy. [13]Hereditary cancer Genetics Group, Vall d'Hebron Institute of Oncology, Vall d'Hebron Hospital Campus, Barcelona, Spain. [14]Department of Medical Oncology, Vall d'Hebron Hospital Universitari, Vall d'Hebron Barcelona Hospital Campus, Barcelona, Spain. [15]Department of Pathology, Landspitali University Hospital, Reykjavik, Iceland. [16]BMC (Biomedical Centre), Faculty of Medicine, University of Iceland, Reykjavik, Iceland. [17]Clinical Genetics Research Lab, Department of Cancer Biology and Genetics, Memorial Sloan Kettering Cancer Center, New York, NY, USA. [18]Department of Clinical Genetics, University of Groningen, University Medical Center Groningen, Groningen, The Netherlands. [19]Department of Clinical Genetics, Maastricht University Medical Center, Maastricht, The Netherlands. [20]Department of Clinical Genetics, Odense University Hospital, Odence C, Denmark. [21]Center for Integrated Oncology (CIO), Faculty of Medicine and University Hospital Cologne, University of Cologne, Cologne, Germany. [22]Center for Molecular Medicine Cologne (CMMC), Faculty of Medicine and University Hospital Cologne, University of Cologne, Cologne, Germany. [23]Center for Familial Breast and Ovarian Cancer, Faculty of Medicine and University Hospital Cologne, University of Cologne, Cologne, Germany. [24]Department of Medicine, Abramson Cancer Center, Perelman School of Medicine at the University of Pennsylvania, Philadelphia, PA, USA. [25]Hereditary Cancer Program, Catalan Institute of Oncology (ICO), ONCOBELL-IDIBELL-IGTP, CIBERONC, Barcelona, Spain. [26]Department of Medicine, Huntsman Cancer Institute, Salt Lake City, UT, USA. [27]SOD Genetica Molecolare, University Hospital, Pisa, Italy. [28]Peter MacCallum Cancer Center, Melbourne, Victoria, Australia. [29]Sir Peter MacCallum Department of Oncology, The University of Melbourne, Melbourne, Victoria, Australia. [30]Departments of Pediatrics and Medicine, Columbia University, New York, NY, USA. [31]Centre for Medical Genetics, Ghent University Hospital, Gent, Belgium. [32]Service de Génétique Biologique, CHRU de Besançon, Besançon, France. [33]Sheffield Clinical Genetics Service, Sheffield Children's Hospital, Sheffield, UK. [34]Gynecologic Oncology, Translational Therapeutics, Department of Obstetrics and Gynecology, Ohio State University Comprehensive Cancer Center, Columbus, OH, USA. [35]Department of Laboratory Medicine and Pathology, Mayo Clinic, Rochester, MN, USA. [36]Department of Clinical Genetics, Fox Chase Cancer Center, Philadelphia, PA, USA. [37]Department of Clinical Genetics, Queen Elizabeth University Hospital, Glasgow, UK. [38]Molecular Oncology Laboratory, CIBERONC, Hospital Clinico San Carlos, IdISSC (Instituto de Investigación Sanitaria del Hospital Clínico San Carlos), Madrid, Spain. [39]Oncogénétique, Institut de Cancérologie de l'Ouest siteRené Gauducheau, Saint Herblain, France. [40]Cancer Genetics and Prevention Program, University of California San Francisco, San Francisco, CA, USA. [41]Area of Clinical and Molecular Genetics, Vall d'Hebron Hospital Universitari, Vall d'Hebron Barcelona Hospital Campus, Barcelona, Spain. [42]Department of Population Sciences, Beckman Research Institute of City of Hope, Duarte, CA, USA. [43]Basser Center for BRCA, Abramson Cancer Center, University of Pennsylvania, Philadelphia, PA, USA. [44]Clinical Genetics Department, St Michael's Hospital, Bristol, UK. [45]Nottingham Clinical Genetics Service, Nottingham University Hospitals NHS Trust, Nottingham, UK. [46]Centre for Cancer Genetic Epidemiology, Department of Oncology, University of Cambridge, Cambridge, UK. [47]Department of Clinical Genetics and Pathology, Laboratory Medicine, Skåne University Hospital, Lund, Sweden. [48]Division of Clinical Genetics, Department of Laboratory Medicine, Lund University, Lund, Sweden. [49]Institute for Medical Informatics, Statistics and Epidemiology, University of Leipzig, Leipzig, Germany. [50]LIFE - Leipzig Research Centre for Civilization Diseases, University of Leipzig, Leipzig, Germany. [51]Division of Evolution and Genomic Sciences, School of Biological Sciences, Faculty of Biology, Medicine and Health, University of Manchester, Manchester Academic Health Science Centre, Manchester, UK. [52]North West Genomics Laboratory Hub, Manchester Centre for Genomic Medicine, St Mary's Hospital, Manchester University NHS Foundation Trust, Manchester Academic Health Science Centre, Manchester, UK. [53]Institute of Medical Genetics and Applied Genomics, University of Tübingen, Tübingen, Germany. [54]Molecular Diagnostics Laboratory, INRASTES, National Centre for Scientific Research 'Demokritos', Athens, Greece. [55]The Susanne Levy Gertner Oncogenetics Unit, Chaim Sheba Medical Center, Ramat Gan, Israel. [56]Sackler Faculty of Medicine, Tel Aviv University, Ramat Aviv, Israel. [57]Clinical Genetics Branch, Division of Cancer Epidemiology and Genetics, National Cancer Institute, Bethesda, MD, USA. [58]Cancer Risk and Prevention Clinic, Dana-Farber Cancer Institute, Boston, MA, USA. [59]Center for Bioinformatics and Functional Genomics and the Cedars Sinai Genomics Core, Cedars-Sinai Medical Center, Los Angeles, CA, USA. [60]Department of Human Genetics, University Würzburg, Würzburg, Germany. [61]Service Régional Oncogénétique Poitou-Charentes, CH Niort, Niort, France. [62]Department of Pathology and Laboratory Medicine, University of Kansas Medical Center, Kansas City, KS, USA. [63]Department of Dermatology, Huntsman Cancer Institute, University of Utah School of Medicine, Salt Lake City, UT, USA. [64]Waukesha Memorial Hospital-Pro Health Care, Waukesha, USA. [65]Molecular Genetics of Breast Cancer, German Cancer Research Center (DKFZ), Heidelberg, Germany. [66]Department of Clinical Genetics, Rigshospitalet, Copenhagen University Hospital, Copenhagen, Denmark.

[67]Institute of Human Genetics, University Hospital Leipzig, Leipzig, Germany. [68]Department of Gynecology and Obstetrics, University Hospital Düsseldorf, Heinrich-Heine University Düsseldorf, Düsseldorf, Germany. [69]Center for Medical Genetics, NorthShore University HealthSystem, Evanston, IL, USA. [70]The University of Chicago Pritzker School of Medicine, Chicago, IL, USA. [71]N.N. Petrov Institute of Oncology, St. Petersburg, Russia. [72]Lombardi Comprehensive Cancer Center, Georgetown University, Washington, DC, USA. [73]Clinical Genetics, Guy's and St Thomas' NHS Foundation Trust, London, UK. [74]Department of Genetics and Pathology, Pomeranian Medical University, Szczecin, Poland. [75]Independent Laboratory of Molecular Biology and Genetic Diagnostics, Pomeranian Medical University, Szczecin, Poland. [76]Parkville Familial Cancer Centre, Peter MacCallum Cancer Center, Melbourne, Victoria, Australia. [77]Faculty of Medicine, Institute of Biomedical Sciences, Dept. Of Human and Medical Genetics, Vilnius University, Vilnius, Lithuania. [78]State Research Institute Centre for Innovative Medicine, Vilnius, Lithuania. [79]Department of Epidemiology & Population Health, Stanford University School of Medicine, Stanford, CA, USA. [80]Department of Medicine, Division of Oncology, Stanford Cancer Institute, Stanford University School of Medicine, Stanford, CA, USA. [81]David Geffen School of Medicine, Department of Obstetrics and Gynecology, University of California at Los Angeles, Los Angeles, CA, USA. [82]Breast and Cancer Genetics Units, The Royal Marsden NHS Foundation Trust, London, UK. [83]Familial Cancer Service, Weatmead Hospital, Wentworthville, New South Wales, Australia. [84]Department of Medical Genetics, University Medical Center, Utrecht, The Netherlands. [85]Hong Kong Hereditary Breast Cancer Family Registry, Hong Kong, China. [86]Department of Surgery, The University of Hong Kong, Hong Kong, China. [87]Department of Surgery and Cancer Genetics Center, Hong Kong Sanatorium and Hospital, Hong Kong, China. [88]Department of Population Health Sciences, Weill Cornell Medicine, New York, NY, USA. [89]Unité de Prévention et d'Epidémiologie Génétique, Centre Léon Bérard, Lyon, France. [90]Department of Clinical Genetics, Aarhus University Hospital, Aarhus N, Denmark. [91]Département de Génétique, CHU de Grenoble, Grenoble, France. [92]Genetic Epidemiology of Cancer team, Inserm U900, Paris, France. [93]Institut Curie, Paris, France. [94]Mines ParisTech, Fontainebleau, France. [95]Magee-Womens Hospital, University of Pittsburgh School of Medicine, Pittsburgh, PA, USA. [96]Département d'Hématologie-Oncologie Médicale, Centre Antoine Lacassagne, Nice, France. [97]Department of Medical Oncology, Erasmus MC Cancer Institute, Rotterdam, The Netherlands. [98]Department of Gynecology and Obstetrics, University of Munich, Campus Großhadern, Munich, Germany. [99]NRG Oncology, Statistics and Data Management Center, Roswell Park Comprehensive Cancer Center, Buffalo, NY, USA. [100]Immunology and Molecular Oncology Unit, Veneto Institute of Oncology IOV - IRCCS, Padua, Italy. [101]Service de Génétique, Institut Curie, Paris, France. [102]Department of Clinical Genetics, Yorkshire Regional Genetics Service, Chapel Allerton Hospital, Leeds, UK. [103]Unité d'oncogénétique, Centre de Lutte Contre le Cancer, Centre Georges-François Leclerc, Dijon, France. [104]Department of Obstetrics and Gynecology, Helsinki University Hospital, University of Helsinki, Helsinki, Finland. [105]Lee Kong Chian School of Medicine, Nanyang Technological University, Singapore, Singapore. [106]Cancer Genetics Service, National Cancer Centre, Singapore, Singapore. [107]Precision Medicine, School of Clinical Sciences at Monash Health, Monash University, Clayton, Victoria, Australia. [108]Department of Clinical Pathology, The University of Melbourne, Melbourne, Victoria, Australia. [109]Latvian Biomedical Research and Study Centre, Riga, Latvia. [110]Clinical Genetics Service, Department of Medicine, Memorial Sloan Kettering Cancer Center, New York, NY, USA. [111]Department of Molecular Genetics, National Institute of Oncology, Budapest, Hungary. [112]Center for Clinical Cancer Genetics, The University of Chicago, Chicago, IL, USA. [113]Familial Cancer Clinical Unit, Human Cancer Genetics Programme, Spanish National Cancer Research Centre (CNIO) and Spanish Network on Rare Diseases (CIBERER), Madrid, Spain. [114]Institute of Medical Genetics and Human Genetics, Charité - Universitätsmedizin Berlin, corporate member of Freie Universität Berlin, Humboldt-Universität zu Berlin and Berlin Institute of Health, Berlin, Germany. [115]Department of Preventive Medicine, Seoul National University College of Medicine, Seoul, Korea. [116]Integrated Major in Innovative Medical Science, Seoul National University College of Medicine, Seoul, South Korea. [117]Cancer Research Institute, Seoul National University, Seoul, Korea. [118]Department of Genetics and Computational Biology, QIMR Berghofer Medical Research Institute, Brisbane, Queensland, Australia. [119]Molecular Diagnostics, Aalborg University Hospital, Aalborg, Denmark. [120]Clinical Cancer Research Center, Aalborg University Hospital, Aalborg, Denmark. [121]Department of Clinical Medicine, Aalborg University, Aalborg, Denmark. [122]Department of Genetics, Portuguese Oncology Institute, Porto, Portugal. [123]Genome Diagnostics Program, IFOM ETS - the AIRC Institute of Molecular Oncology, Milan, Italy. [124]Unit of Molecular Bases of Genetic Risk and Genetic Testing, Department of Research, Fondazione IRCCS Istituto Nazionale dei Tumori (INT), Milan, Italy. [125]Division of Gynaecology and Obstetrics, Klinikum rechts der Isar der Technischen Universität München, Munich, Germany. [126]Clinical Genetics, Karolinska Institutet, Stockholm, Sweden. [127]Division of Gynecologic Oncology, NorthShore University HealthSystem, University of Chicago, Evanston, IL, USA. [128]Department of Clinical Genetics, University Hospital of Southern Denmark, Vejle Hospital, Vejle, Denmark. [129]Department of Pathology, The Netherlands Cancer Institute - Antoni van Leeuwenhoek Hospital, Amsterdam, The Netherlands. [130]Center for Genomic Medicine, Rigshospitalet, Copenhagen, Denmark. [131]Department of Clinical Medicine, University of Copenhagen, Copenhagen, Denmark. [132]West Midlands Regional Genetics Service, Birmingham Women's Hospital Healthcare NHS Trust, Birmingham, UK. [133]Department of Internal Medicine, Division of Medical Oncology, University of Kansas Medical Center, Westwood, KS, USA. [134]Princess Anne Hospital, Southampton, UK. [135]Genomics Center, Centre Hospitalier Universitaire de Québec – Université Laval Research Center, Québec City, QC, Canada. [136]Dept of OB/GYN and Comprehensive Cancer Center, Medical University of Vienna, Vienna, Austria. [137]Medical Genetics Unit, St George's, University of London, London, UK. [138]Institute of Human Genetics, Hannover Medical School, Hannover, Germany. [139]Department of Tumour Biology, INSERM U830, Paris, France. [140]Université Paris Cité, Paris, France. [141]Institute of Human Genetics, University Hospital Heidelberg, Heidelberg, Germany. [142]Biomedical Sciences Institute (ICBAS), University of Porto, Porto, Portugal. [143]Breast Cancer Research Programme, Cancer Research Malaysia, Subang Jaya, Selangor, Malaysia. [144]Department of Surgery, Faculty of Medicine, University of Malaya, Kuala Lumpur, Malaysia. [145]Department of Medicine, Magee-Womens Hospital, University of Pittsburgh School of Medicine, Pittsburgh, PA, USA. [146]Program in Cancer Genetics, Departments of Human Genetics and Oncology, McGill University, Montréal, QC, Canada. [147]Department of Medical Genetics, National Institute for Health Research Cambridge Biomedical Research Centre, University of Cambridge, Cambridge, UK. [148]Department of Cancer Biology and Genetics, The Ohio State University, Columbus, OH, USA. [149]Department of medicine, University Of Melbourne, Melbourne, Victoria, Australia. [150]South East Thames Regional Genetics Service, Guy's and St Thomas' NHS Foundation Trust, London, UK. [151]Department of Medical Oncology, Beth Israel Deaconess Medical Center, Boston, MA, USA. [152]Department of Clinical Genetics, VU University Medical Center, Amsterdam, The Netherlands. [153]Department of Genetics, University of Pretoria, Arcadia, South Africa. [154]Centro de Investigación en Red de Enfermedades Raras (CIBERER), Madrid, Spain. [155]Fundación Pública Galega de Medicina Xenómica, Santiago de Compostela, Spain. [156]Instituto de Investigación Sanitaria de Santiago de Compostela (IDIS), Complejo Hospitalario Universitario de Santiago, SERGAS, Santiago de Compostela, Spain. [157]Division of Functional onco-genomics and genetics, Centro di Riferimento Oncologico di Aviano (CRO), IRCCS, Aviano, Italy. [158]Oxford Regional Genetics Service, Churchill Hospital, Oxford, UK. [159]Latin American School of Oncology, Tuxtla Gutiérrez, Chiapas, Mexico. [160]Radboud University Medical Center, Nijmegen, Netherlands. *Lists of authors and their affiliations appear at the end of the paper. ✉email: logan.walker@otago.ac.nz

## GEMO Study Collaborators

Marie-Agnès Collonge-Rame[32], Capucine Delnatte[39], Paul Gesta[61], Christine Lasset[89], Clémentine Legrand[91], Fabienne Lesueur[92,93,94], Véronique Mari[96], Noura Mebirouk[92,93,94], Emmanuelle Mouret-Fourme[101], Sophie Nambot[103] & Dominique Stoppa-Lyonnet[101,139,140]

## EMBRACE Collaborators

Jackie Cook[33], Rosemarie Davidson[37], Alan Donaldson[44], Jacqueline Eason[45], D. Gareth Evans[51,52], Debra Frost[4], Louise Izatt[73], Zoe Kemp[82], Fiona Lalloo[88], Hannah Musgrave[102], Saba Sharif[132], Lucy E. Side[134], Katie Snape[137], Vishakha Tripathi[150] & Lisa Walker[158]

## SWE-BRCA Investigators

Hans Ehrencrona[47,48] & Johanna Rantala[126]

## kConFab Investigators

Georgia Chenevix-Trench[118] & Amanda B. Spurdle[118]

## HEBON Investigators

Lieke Berger[18], Marinus J. Blok[19], Marco Koudijs[84], John W. M. Martens[97], Efraim H. Rosenberg[129], Klaartje van Engelen[152] & Marijke R. Wevers[160]

