## [Peer Review File · Communications Biology]

Reviewers' comments:

Reviewer #1 (Remarks to the Author):

This manuscript describes associations between CNVs and breast cancer risk in pathogenic carriers of BRCA1 or BRCA2. The study is a nice contribution to the literature, though the findings may have been a bit overinterpreted. In addition, the manuscript would benefit from addressing the following:

Abstract

1. It would be useful for the authors to show the calculation whereby a significance threshold of 0.01 is appropriate.
2. The authors should avoid the use of causal language.
3. Were the findings regarding pathogenic deletions in BRCA1 discovered in the population of BRCA1 pathogenic variant carriers (i.e., excluding BRCA2 pathogenic variant carriers)?

Introduction

4. The statement that "single nucleotide variants only account for a fraction of heritable variation in risk in BRCA1/2 pathogenic variant carriers" requires a citation. And what is the fraction?
5. The Introduction otherwise does a nice job of motivating the study.

Results

6. I'm not especially familiar with the available tools to call CNVs, but it strikes me that some of the sensitivities and specificities are rather low. Can the authors comment (in the Discussion) on the impact for their analyses?
7. Again, the authors should offer justification for their choice of significance threshold.
8. The hazard ratio for breast cancer risk among BRCA2 duplication carriers versus carriers of other pathogenic variant types is larger than that for BRCA1 duplications versus non-duplication pathogenic variants. As such, I don't know that it's reasonable to suggest that the latter result is suggestive and that there is no evidence of an association for the former. There is a clear issue of sample size, so any interpretation of the results should keep that in mind.
9. Much of the material in the first paragraph of the "Identification of SULT1A1 as a candidate modifier gene" section would be better suited for the Discussion.

Discussion

10. Given what I consider to be a rather liberal significance threshold and experiments with mixed results, I encourage the authors to soften their conclusions throughout the Discussion.
11. In the Introduction, the authors mention prior studies that have evaluated CNVs in the general population. How do the authors' results compare? Do they find associations that only appear in individuals with BRCA1/BRCA1 pathogenic variants? Are there associations that they do NOT find that they might have expected?
12. An additional limitation is the lack of a replication cohort.

Methods

13. I'd like to see better characterization of the study participants (either when described in the Methods or in the Results). Mean age at enrollment? Any information about family history of breast cancer? Etc.
14. Did the authors consider analyses that combine pathogenic carriers of BRCA1 or BRCA2?
15. The Methods need to make clearer the analyses that achieved RRs versus the analyses that achieved HRs. They should also mention covariates.
16. I'm not qualified to comment on the experimental methods.

Reviewer #2 (Remarks to the Author):

Hakkaart et al present a study entitled "Copy Number Variants as Modifiers of Breast Cancer Risk for BRCA1/BRCA2 Pathogenic Variant", looking at CNV modifiers of breast cancer risk in germline BRCA1/2 mutation carriers. Although a potentially interesting study, and the authors should be commended for their functional work, there are significant issues with the analyses and

conclusions and thus is not suitable to publish in its current state in Communications Biology.

1. Unclear where the p value for significance comes from- $p < 0.01$ – which is not corrected for multiple comparisons. In fact, no CNV regions are significantly associated with breast cancer risk when considering q-values corrected for multiple testing (q-values are listed in the supplement but never mentioned in results or discussion. CNVs in 16,395 unique gene regions were tested; $p < 0.01$ is an inappropriate threshold for statistical significance. The authors appropriately state in the introduction, “CNVs previously have been shown to be modifiers of hereditary breast cancer risk. In a genome-wide association analysis of CNVs in 2500 BRCA1 pathogenic variant carriers, 52 gene loci were associated (unadjusted $p < 0.05$) with breast cancer risk (Walker, Marquart, et al., 2017). Although no variant reached the widely-adopted genome-wide statistical significance threshold ($p < 5 \times 10^{-8}$) and the study sample size was relatively small, the specific genes disrupted by CNVs had plausible biological consequences regarding cancer development. These data suggested that CNVs are an important modifier of hereditary breast cancer risk and highlighted the need for larger and more comprehensive CNV studies.” The data presented here seems to similarly show a trend towards significance for genes that are plausible to be involved in breast cancer risk but even with a substantially larger study, none of these CNVs reach statistical significance after correction for multiple comparisons. At a minimum the p value threshold needs to be explained and the modest associations put into context. With that said, the SULT1A1 functional data is suggestive of deletions in this gene modifying cancer risk, although these data are subtle and are only presented for one cell line.

2. The authors spend a substantial amount of space in the manuscript evaluating their CNV calling. While they should be in part commended for these efforts, the general lack of specificity and sensitivity in the supplemental tables is worrisome (especially with the lack of validating TERT, LSP1, etc.) and makes one question the modest associations presented even further. Further, the Walker et al study pointed to by the authors identifies a completely different list of breast cancer associated CNVs which at the very least needs to be put into context, but also raises questions about the calling accuracy. This much smaller size study had CNVs that were in the same range of modest statistical significances which could suggest many CNVs are being missed in these analyses. Finally, only a handful of candidate CNVs were validated (ex. SULT1A1 deletion was validated in 8 out of 200+ samples).

3. The results section labeled “CNVs associated with breast cancer risk” is very difficult to follow.

4. Figure 3 is labeled incorrectly as there is no MMC treatment indicated.

5. The functional data is the strongest aspect of the paper, although it is unclear why they prioritized this gene for functional validation. Further, wonder if they could test the hypothesis they make in the discussion – “The mechanism by which CNV deletions overlapping SULT1A1 were associated with lower BRCA1-associated breast cancer risk may be linked to the production of potentially toxic catechol oestrogens by the Cytochromes P450 (CYP) enzymes....” Also, unclear why the authors chose to use siRNA to kd SULT1A1 when clearly they have the ability to utilize CRISPR and even shRNA would even afford the opportunity to generate stable isogenic clones. Finally, the relevance of the chosen BRCA1 mutation to model with SULT1A1 kd is unclear? Is this a common BRCA1 germline mutation? Why not model several background mutations? Would be best to model this in normal breast cells not an already transformed cell line.

Reviewer #1 (Remarks to the Author):

Abstract

- 1. It would be useful for the authors to show the calculation whereby a significance threshold of 0.01 is appropriate.

Authors' response: Nominal significance without multiple testing adjustment is $p \leq 0.05$ by general convention. Our focus was to identify 'top-hits' to prioritise *in silico* and functional analyses, and to do this we implemented an arbitrary cut-off at the more stringent $P < 0.01$. Our study identified more than 350 putative CNV loci that were associated with breast cancer risk in *BRCA1* and/or *BRCA2* pathogenic variant carriers at $P < 0.05$. To clarify this in the manuscript we have added the sentence (Line 384) – “*To prioritise genes for in silico and functional analyses, we opted to implement the $p < 0.01$ threshold to narrow the list of candidate gene loci*”.

- 2. The authors should avoid the use of causal language.

Authors' response: We have made the following correction in the abstract – (Line 288) “*Notably, pathogenic deletions in BRCA1 were associated with increased breast cancer risk...*”

- 3. Were the findings regarding pathogenic deletions in *BRCA1* discovered in the population of *BRCA1* pathogenic variant carriers (i.e., excluding *BRCA2* pathogenic variant carriers)?

Authors' response: Yes. This is especially highlighted in Table 1, which notes that the analysis was conducted separately for *BRCA1* and *BRCA2* carriers

Introduction

- 4. The statement that “single nucleotide variants only account for a fraction of heritable variation in risk in *BRCA1/2* pathogenic variant carriers” requires a citation. And what is the fraction?

Authors' response: We have now provided more detail (Line 313) – “*However, the identified single nucleotide variant modifiers ~~only~~ account for ~~a fraction~~ less than 10% of heritable variation in risk in *BRCA1/2* pathogenic variant carriers (Milne et al., 2017).*”

- 5. The Introduction otherwise does a nice job of motivating the study.

Authors' response: No response required

Results

- 6. I'm not especially familiar with the available tools to call CNVs, but it strikes me that some of the sensitivities and specificities are rather low. Can the authors comment (in the Discussion) on the impact for their analyses?

Authors' response: We have added the following comments in the Discussion – (Line 487)

“*Consistent with observations from the human CNV map, we validated positive CNV calls overlapping the *SULT1A1* gene, and revealed false positive CNV calls at two candidate modifier gene regions (*LSP1* and *TERT*).*”

(Line 579) “*Furthermore, CNV calling algorithms have limitations which lead to false CNV calls, thus highlighting the importance of using ancillary data to prioritise regions for downstream analyses.*”

- 7. Again, the authors should offer justification for their choice of significance threshold.

Authors' response: See response above

- 8. The hazard ratio for breast cancer risk among *BRCA2* duplication carriers versus carriers of other pathogenic variant types is larger than that for *BRCA1* duplications versus non-duplication pathogenic variants. As such, I don't know that it's reasonable to suggest that the latter result is suggestive and that there is no evidence of an association for the former. There is a clear issue of sample size, so any interpretation of the results should keep that in mind.

Authors' response: We agree with the reviewer that there may be a statistical power issue in relation to CNVs overlapping *BRCA2*. We have made the following addition to the results section – (Line 400) “*There was no significant evidence of increased in risk among BRCA2 duplication carriers versus carriers of other pathogenic variant types (HR=1.52, 95%CI=0.61-3.77, p=0.39), but results were less certain given the smaller sample size and wide confidence intervals.*”

- 9. Much of the material in the first paragraph of the “Identification of *SULT1A1* as a candidate modifier gene” section would be better suited for the Discussion.

Authors' response: We have moved most of that first paragraph to the discussion section.

- 10. Given what I consider to be a rather liberal significance threshold and experiments with mixed results, I encourage the authors to soften their conclusions throughout the Discussion.

Authors' response: We have now softened our conclusions and clarified that the associations were found using unadjusted $P < 0.01$.

(Line 485) “*We identified putative CNVs in up to 31 putative gene regions that were associated (unadjusted $P < 0.01$) with breast cancer risk for BRCA1/2 pathogenic variant carriers, with CNVs at 15 of these regions present in a human CNV map (Zarrei et al., 2015). Consistent with observations from the human CNV map, we validated positive CNV calls overlapping the *SULT1A1* gene, and revealed false positive CNV calls at two candidate modifier gene regions (*LSP1* and *TERT*).*”

- 11. In the Introduction, the authors mention prior studies that have evaluated CNVs in the general population. How do the authors' results compare? Do they find associations that only appear in individuals with *BRCA1/BRCA1* pathogenic variants? Are there associations that they do NOT find that they might have expected?

Authors' response: As noted above we used a population based human CNV map to provide confidence in the CNV calls. This study is the only CNV study of *BRCA1/BRCA2* pathogenic variant carriers that has utilised the custom OncoArray and it is the largest to date. We did compare our results with established modifier loci from previous SNP association studies conducted by the CIMBA consortium. As described in the manuscript (Line 407), our results indicated that “*Putative deletions overlapping TERT and duplications overlapping LSP1, two loci previously shown to be associated with breast cancer risk for BRCA1 (TERT locus) and BRCA2 (TERT and LSP1 loci)*”. However, these rare CNVs did not appear in the Zarrei et al human CNV map and 5/7 CNVs did not validate using qRT-PCR. Only one gene locus (*STK11*) listed in Supplementary Tables S5-S8 was associated with breast cancer for both *BRCA1* and *BRCA2* pathogenic variant carriers (Line 405).

- 12. An additional limitation is the lack of a replication cohort.

Authors' response: We agree that the results would be strengthened through replication but this would be a significant undertaking requiring extended collaboration, and separate to the current study. For example, ongoing large scale GWAS in additional carriers is underway with the CONFLUENCE project (<https://dceg.cancer.gov/research/cancer-types/breast-cancer/confluence-project>) but data will not be reported for another 2-3 years.

- 13. I'd like to see better characterization of the study participants (either when described in the Methods or in the Results). Mean age at enrollment? Any information about family history of breast cancer? Etc.

Authors' response: We have summarised these data in Supplementary Table 12 and referred to this table in the Study Cohort section of the Methods.

-14. Did the authors consider analyses that combine pathogenic carriers of *BRCA1* or *BRCA2*?

Authors' response: Although this could potentially increase power, *BRCA1* and *BRCA2* are distinct genes with distinct age-specific breast cancer incidence patterns. Moreover, carriers of pathogenic variants in *BRCA1* and *BRCA2* develop distinct breast cancer subtypes and it has been demonstrated through GWAS that the patterns of association of SNP modifiers are different in *BRCA1* and *BRCA2* carriers. SNPs that modify breast cancer risk for *BRCA1* carriers tend to be SNPs which are associated ER-negative breast cancer risk in the population (the predominant breast cancer subtype in *BRCA1* carriers), whereas the SNPs which modify breast cancer risk for *BRCA2* carriers, tend to be the SNPs associated with overall or ER-positive breast cancer in the population (Milne et al, 2017 Nature Genetics, 49(12), 1767–1778; Zhang et al Nature Genetics, 52(6), 572–581).

- 15. The Methods need to make clearer the analyses that achieved RRs versus the analyses that achieved HRs. They should also mention covariates.

Authors' response: This has now been clarified in the methods (Lines 650-670) and amended in the Supplementary Tables 5-8

Reviewer #2 (Remarks to the Author):

- 1. Unclear where the p value for significance comes from- $p < 0.01$ – which is not corrected for multiple comparisons. In fact, no CNV regions are significantly associated with breast cancer risk when considering q-values corrected for multiple testing (q-values are listed in the supplement but never mentioned in results or discussion. CNVs in 16,395 unique gene regions were tested; $p < 0.01$ is an inappropriate threshold for statistical significance.... At a minimum the p value threshold needs to be explained and the modest associations put into context. With that said, the SULT1A1 functional data is suggestive of deletions in this gene modifying cancer risk, although these data are subtle and are only presented for one cell line.

Authors' response: Please note our comment to Reviewer 1 to a very similar question. We recognise that despite the relatively large cohort study, power to confirm CNV loci with a low MAF was a limitation, and we specifically noted this in the discussion (Line 574).

- 2. The authors spend a substantial amount of space in the manuscript evaluating their CNV calling. While they should be in part commended for these efforts, the general lack of specificity and sensitivity in the supplemental tables is worrisome (especially with the lack of validating TERT, LSP1, etc.) and makes one question the modest associations presented even further. Further, the Walker et al study pointed to by the authors identifies a completely different list of breast cancer associated CNVs which at the very least needs to be put into context, but also raises questions about the calling accuracy. This much smaller size study had CNVs that were in the same range of modest statistical significances which could suggest many CNVs are being missed in these analyses.

Authors' response: It is important to note that all CNVs listed in Supplementary Tables 5-8 were compared with a human CNV map to ascertain the level of confidence in the CNV calling. We have now summarised these comparisons in the results (Line 385) "*Putative CNVs at Thirty-one 31 gene regions were associated ($p < 0.01$) with breast cancer risk (Supplementary Table 5-8), however for 16 of these 31 regions, the proportion of unique CNVs represented in a published human CNV map (Zarrei et al., 2015) was less than 95%.*

It is also important to reiterate that data from our CNV calling detected an association between deletions overlapping *BRCA1* and breast cancer risk. We subsequently validated this finding from our study cohort using data that included clinically diagnosed *BRCA1* variants.

The custom Oncoarray used in this study differs significantly in probe design from the Human610-Quad BeadChip previously used in Walker et al EJHG (2017), 1–7. It is therefore not surprising that different gene loci were identified between the two studies.

- Finally, only a handful of candidate CNVs were validated (ex. SULT1A1 deletion was validated in 8 out of 200+ samples).

Authors' response: DNA for orthogonal technologies was available from a limited number of samples. Please also note our comments above regarding CNV calling and validation.

- 3. The results section labeled “CNVs associated with breast cancer risk” is very difficult to follow.

Authors' response: Our amendments have hopefully addressed this issue – (Line 389) “*Deletions overlapping BRCA1 were associated with increased breast cancer risk (hazard ratio (HR)=1.29, 95%CI=1.13-1.49, p=1.98x10⁻⁴) (Supplementary Table 5) for BRCA1 pathogenic variant carriers. This result was explored further as the analysis did not directly compare the effect of BRCA1 deletions carriers and BRCA1 non-deletion pathogenic variants carriers. Clinically diagnosed variants for BRCA1 and BRCA2 carriers were categorised by type (deletions, duplications, and small variants [i.e. nonsense, missense, frame shift, Indel, and splice site]). Assessing the HRs for ~~BRCA1 CNV versus non-CNV pathogenic variants, or BRCA2 CNV versus non-CNV pathogenic variants~~ separately for BRCA1 and BRCA2 confirmed that breast cancer risk was increased for BRCA1 deletions (HR=1.21, 95%CI=1.09-1.35) but not BRCA2 deletions (Table 1, Supplementary Table 6).”*

- 4. Figure 3 is labeled incorrectly as there is no MMC treatment indicated.

Authors' response: This has been corrected.

- 5. The functional data is the strongest aspect of the paper, although it is unclear why they prioritized this gene for functional validation. Further, wonder if they could test the hypothesis they make in the discussion – “The mechanism by which CNV deletions overlapping SULT1A1 were associated with lower BRCA1-associated breast cancer risk may be linked to the production of potentially toxic catechol oestrogens by the Cytochromes P450 (CYP) enzymes....”

Authors' response: We outline our rationale for choosing the *SULT1A1* gene in the results section (Line 418) “*Identification of SULT1A1 as a candidate modifier gene*”. Firstly, *SULT1A1* deletions were associated with decreased breast cancer risk in *BRCA1* carriers. Secondly, *SULT1A1* plays an important role in the metabolism, bioactivation, and detoxification of carcinogens, medications, and steroid hormones. Thirdly, overlapping *SULT1A1* CNVs had a population frequency above 1%.

We agree that that the link between *SULT1A1*, *BRCA1* pathogenic variants and oestrogen metabolism is worth investigating, however such undertaking goes beyond the scope of the current study. We have made the following addition to the Discussion (Line 552) – “*These results further suggest that the balance between the generation of catecholestrogens and catecholesterogen sulfation may be an important mechanism for modulating breast cancer risk and worthy of future investigation.*”

- Also, unclear why the authors chose to use siRNA to kd *SULT1A1* when clearly they have the ability to utilize CRISPR and even shRNA would even afford the opportunity to generate stable isogenic clones. Finally, the relevance of the chosen *BRCA1* mutation to model with *SULT1A1* kd is unclear? Is this a common *BRCA1* germline mutation? Why not model several background mutations? Would be best to model this in normal breast cells not an already transformed cell line.

Authors' response: *SULT1A1* was shown to be a dosage sensitive gene with expression correlating to gene copy number. We therefore chose the siRNA knockdown approach to model the impact of the deletion on *SULT1A1* expression. The ‘normal’ MCF10A cell line was available to us in the laboratory but this cell line expressed no quantifiable *SULT1A1* RNA. Modelling numerous CRISPR generated *BRCA1* pathogenic variants would have been a significant undertaking and we did not have the resources to extend the study further.

Reviewers' comments:

Reviewer #1 (Remarks to the Author):

The authors adequately addressed my initial feedback.

Reviewer #2 (Remarks to the Author):

Our original concern over lack of correction for multiple comparisons remains and there was not an adequate explanation given for this deviation from common practice with GWAS studies (remark 1). No CNVs reached the genome-wide significance threshold and thus it cannot be stated that there are CNVs associated with BC risk. In general, our remarks 2 and 5 were also not adequately addressed or at the most, incompletely addressed. As such, we cannot recommend publication in the current state.

Reviewer #2 (Remarks to the Author):

Our original concern over lack of correction for multiple comparisons remains and there was not an adequate explanation given for this deviation from common practice with GWAS studies (remark 1). No CNVs reached the genome-wide significance threshold and thus it cannot be stated that there are CNVs associated with BC risk.

Authors' response: To address the reviewer's concerns, we have now included yet additional text around P values and reworded key points in the Abstract, Results and Discussion to refer to risk estimates (and not claims of genome-wide statistical significance). Please note that in the present study we are evaluating CNVs, as opposed to SNPs, so the usual genome-wide significance level of 5×10^{-8} is not applicable in this case. A more appropriate adjustment would be on the basis of the gene-regions considered for each association analysis in this study. Our changes are as follows:

- ABSTRACT

Line 281 – *“We used these results to prioritise a candidate breast cancer risk-modifier gene for laboratory analysis and biological validation identified 31 genomic loci associated ($p < 0.01$) with the risk of breast cancer development.”*

Line 282 – *“Notably, the HR for deletions in BRCA1 suggested an elevated breast cancer risk estimate pathogenic deletions in BRCA1 were associated with increased breast cancer risk (hazard ratio (HR)=1.21, 95% confidence interval (95% CI)=1.09-1.35) compared with non-CNV pathogenic variants.”*

Line 284 – *“In contrast, deletions overlapping SULT1A1 suggested a decreased ~~were associated~~ with reduced breast cancer risk (HR=0.73, 95% CI 0.59-0.91) in BRCA1 pathogenic variant carriers.”*

- INTRODUCTION

Line 325 – *“Although no variant reached the widely-adopted genome-wide statistical significance threshold applied for SNP-centric GWAS ($p < 5 \times 10^{-8}$) ...”*

- RESULTS

Line 377 – *“Prioritization of candidate breast cancer CNV risk loci ~~associated with breast cancer risk~~”*

Line 378 – *“To prioritise genes for in silico and functional analyses, we opted to implement the selected those candidate gene loci with $p < 0.01$ from retrospective likelihood analysis, effectively restricting hazard ratios to > 1.25 and < 0.75 (Supplementary Tables 5-8) threshold to narrow the list of candidate gene loci. Putative CNVs at 31 gene regions ~~were associated with breast cancer risk (Supplementary Table 5-8)~~ passed this threshold. ~~however fo~~ For 16 of these 31 regions, the proportion of unique CNVs represented in a published human CNV map (Zarrei et al., 2015) was less than 95%. Although none of the CNV regions passed significance thresholds when adjusted for multiple hypothesis testing (See Methods; deletions in BRCA1 carriers - $p \leq 8 \times 10^{-6}$; duplications in BRCA1 carriers - $p \leq 5 \times 10^{-6}$; deletions in BRCA2 carriers - $p \leq 1 \times 10^{-5}$; and duplications in BRCA2 carriers - $p \leq 6 \times 10^{-6}$), we used these results to prioritise a candidate risk-modifier gene for laboratory analysis and biological validation.”*

Line 388 – *“Deletions overlapping BRCA1 ~~were nominally associated with~~ increased breast cancer risk ...”*

Line 393 – *“Assessing the HRs for CNV versus non-CNV pathogenic variants, separately for BRCA1 and BRCA2 ~~confirmed suggested elevated that~~ breast cancer risk ~~was increased for~~ BRCA1 deletions (HR=1.21, 95%CI=1.09-1.35) but not BRCA2 deletions (Table 1, Supplementary Table*

9). These results remained similar ~~to~~ after excluding missense variant carriers from the analysis (Supplementary Table 9). Interestingly, HRs were elevated estimates for BRCA1 duplications versus non-duplication pathogenic variants (deletions were excluded) for BRCA1 suggested an increased risk of breast cancer for duplication carriers (HR=1.21, 95%CI=0.99-1.48; p=0.066), although this association was not statistically significant (p=0.66). There was no significant evidence of increased risk among BRCA2 duplication carriers versus carriers of other pathogenic variant types (HR=1.52, 95%CI=0.61-3.77, p=0.39); however, results for BRCA2 were less definitive given the smaller sample size and wide confidence intervals.

Line 402 – “Putative duplications overlapping the breast cancer tumour suppressor gene STK11 were associated with suggested decreased risk of breast cancer in our study for both BRCA1 carriers (HR=0.49, 95%CI 0.29-0.81, p=5.4x10⁻³) and BRCA2 carriers (HR=0.44, 95%CI 0.22-0.88, p=9.2x10⁻³). Putative deletions overlapping TERT and duplications overlapping LSP1, two loci previously shown to be associated with breast cancer risk for BRCA1 (TERT locus) and BRCA2 (TERT and LSP1 loci) pathogenic variant carriers from SNP-based studies (Antoniou et al., 2009; Bojesen et al., 2013), were also associated with suggested increased risk (HR=1.92, 95%CI=1.06-3.46, p=6.0x10⁻³) and decreased risk (HR=0.13, 95%CI=0.04-0.45, ...”

Line 417 – “CNV loci associated suggested to modify with breast cancer risk estimates in BRCA1/2 pathogenic variant carriers ...”

Line 422 – “In our study, CNV deletions overlapping SULT1A1 were identified in 1.7% of BRCA1 pathogenic variant carriers and were associated they suggested with a decreased breast cancer risk HR (HR=0.73, 95%CI=0.59-0.91, p=9.1x10⁻³).”

Line 446 – “As deletions overlapping SULT1A1 were associated with suggested to decrease breast cancer risk...”

- DISCUSSION

Line 485 – “Although none of the CNV regions passed significance thresholds when adjusted for multiple hypothesis testing, we used these results to prioritise a candidate risk-modifier gene for laboratory analysis and biological validation”

Line 490 – “CNV deletions overlapping the lead candidate modifier SULT1A1 were associated showed with decreased...”

Line 575 – “Despite this being the relatively largest sample size of BRCA1 and BRCA2 pathogenic variant carriers available to date, the low frequency of CNVs results in limited the power to for detecting significant associations after adjusting for multiple comparisons. As a result, a nominal screening threshold of 0.01 was used which is arbitrary and is therefore a limitation of the study. Nevertheless, this is the largest extant dataset available for examining genetic modifiers of BRCA1 and BRCA2 related risk.”

Line 663 - “Conservative significance thresholds were based on the number of effective tests in this gene-centric CNV study. After excluding gene regions with no overlapping CNVs, thresholds were as follows: deletions in BRCA1 carriers - $p \leq 0.05/6551 = 8 \times 10^{-6}$; duplications in BRCA1 carriers - $p \leq 0.05/10240 = 5 \times 10^{-6}$; deletions in BRCA2 carriers - $p \leq 0.05/5094 = 1 \times 10^{-5}$; and duplications in BRCA2 carriers - $p \leq 0.05/8469 = 6 \times 10^{-6}$.”

- SUPPLEMENTARY INFORMATION

Q-values have been removed from Supplementary Tables 5-8

In general, our remarks 2 and 5 were also not adequately addressed or at the most, incompletely addressed. As such, we cannot recommend publication in the current state.

Authors' response: We have added an extra reference to justify SULT1A1 as a candidate risk modifier based on its biological function and note, as for Reviewer #1, further functional analysis is outside the scope of this study.

Line 542 – *“Indeed, the SULT1A1 substrate, 2-MeOE2, has previously been proposed as a potential preventative agent for breast cancer (Zhu et al., 1998).”*

REVIEWERS' COMMENTS:

Reviewer #2 (Remarks to the Author):

The authors have adequately addressed our comments.